# The RNA-binding protein Msi2 regulates autophagy during myogenic differentiation

Ruochong Wang[1,2] , Futaba Kato[1], Rio Yasui Watson[1,2], Aaron M Beedle[3,5] , Jarrod A Call[4], Yugo Tsunoda[1], Takeshi Noda[1], Takaho Tsuchiya[6], Makoto Kashima[7,8], Ayuna Hattori[1,2], Takahiro Ito[1,2]

**Skeletal muscle development is a highly ordered process orchestrated transcriptionally by the myogenic regulatory factors. However, the downstream molecular mechanisms of myogenic regulatory factor functions in myogenesis are not fully understood. Here, we identified the RNA-binding protein Musashi2 (Msi2) as a myogenin target gene and a post-transcriptional regulator of myoblast differentiation. Msi2 knockdown in murine myoblasts blocked differentiation without affecting the expression of MyoD or myogenin. Msi2 overexpression was also sufficient to promote myoblast differentiation and myocyte fusion. Msi2 loss attenuated autophagosome formation via down-regulation of the autophagic protein MAPL1LC3/ATG8 (LC3) at the early phase of myoblast differentiation. Moreover, forced activation of autophagy effectively suppressed the differentiation defects incurred by Msi2 loss. Consistent with its functions in myoblasts in vitro, mice deficient for Msi2 exhibited smaller limb skeletal muscles, poorer exercise performance, and muscle fiber–type switching in vivo. Collectively, our study demonstrates that Msi2 is a novel regulator of mammalian myogenesis and establishes a new functional link between muscular development and autophagy regulation.**

## Introduction

Formation of functioning skeletal muscles, that is, myogenesis, is a highly ordered process involving multiple steps of cell fate decisions (Bentzinger et al, 2012). Skeletal muscles are made of myofiber bundles, which are composed of myotubes. Myotubes are unique in that they are generated via fusion of mature myocytes differentiated from myoblasts. Myoblasts are myogenic progenitor cells and can proliferate extensively. Proliferating myoblasts undergo cell fate determination and exit the active cell cycle to differentiate into myocytes. The process of myogenic cell fate determination is regulated by a family of four closely related basic helix–loop–helix transcription factors known as myogenic regulatory factors (MRFs), namely, MyoD, Myf5, Mrf4, and myogenin (Myog) (Wagers & Conboy, 2005; Zammit, 2017). MRFs bind to a specific DNA sequence, the CANNTG E-box cis-elements, located in the regulatory regions of their target genes. Three of the MRFs have overlapping and redundant functions in vivo, and thus, mice singly deficient for Myf5, MyoD, or MRF4 showed no overt abnormalities in myogenesis (Rudnicki et al, 1992; Rawls et al, Dev Biol 1995; Zhang et al, 1995). In contrast, single-gene knockout of *Myog* results in differentiation arrest of myoblasts and defective myofiber formation, indicating that Myog has a distinct and essential function in the myogenic pathway, which cannot be compensated by MyoD and Myf5 (Hasty et al, 1993; Nabeshima et al, 1993). It remains elusive, however, how Myog regulates myogenesis and thus the skeletal muscle cell fates.

In this study, we identified Musashi2 (Msi2) as an essential mediator of Myog function and autophagy in myogenic cell fate decisions. Msi2 is an RNA-binding protein (RBP) of the Musashi family that regulates cell fate decisions in various tissue systems (Okano et al, 2002; Horisawa et al, 2010; Hattori et al, 2016). In mammalian species, two Msi-coding genes, namely, *Msi1* and *Msi2*, are present. The two genes encode structurally similar proteins with two highly conserved RNA Recognition Motifs (RRMs) for binding to their specific mRNA targets, whereas the genes exhibit distinct expression patterns (Sakakibara et al, 2001). Msi1 has been extensively studied for its expression and function in stem cells of various adult tissues, such as brain and intestine, as well as breast and colon cancers (Sakakibara et al, 2002; Potten et al, 2003; Wang et al, 2010; Chiou et al, 2017). Although both Msi1 and Msi2 are expressed in brain and testis, Msi2 is dominant in the hematopoietic tissues and functionally essential for normal hematopoietic stem cells and leukemia stem cells in myeloid leukemia (Ito et al, 2010; Park et al, 2014; Hattori et al, 2017). Msi2 has also been implicated in non-hematopoietic tumors as well such as breast cancer

[1]Institute for Life and Medical Sciences, Kyoto University, Kyoto, Japan  [2]Department of Biochemistry and Molecular Biology, The University of Georgia, Athens, GA, USA  [3]Department of Pharmaceutical and Biomedical Sciences, The University of Georgia, Athens, GA, USA  [4]Department of Physiology & Pharmacology, The University of Georgia, Athens, GA, USA  [5]Department of Pharmaceutical Sciences, SUNY Binghamton University, New York, NY, USA  [6]Bioinformatics Laboratory, Institute of Medicine, and Center for Artificial Intelligence Research, University of Tsukuba, Tsukuba, Japan  [7]College of Science and Engineering, Aoyama Gakuin University, Kanagawa, Japan  [8]Department of Molecular Biology, Faculty of Science, Toho University, Chiba, Japan

Correspondence: takahiro.ito@infront.kyoto-u.ac.jp; ito@bmb.uga.edu
Ruochong Wang's present address is Department of Cell Biology, Emory University School of Medicine, Atlanta, GA, USA

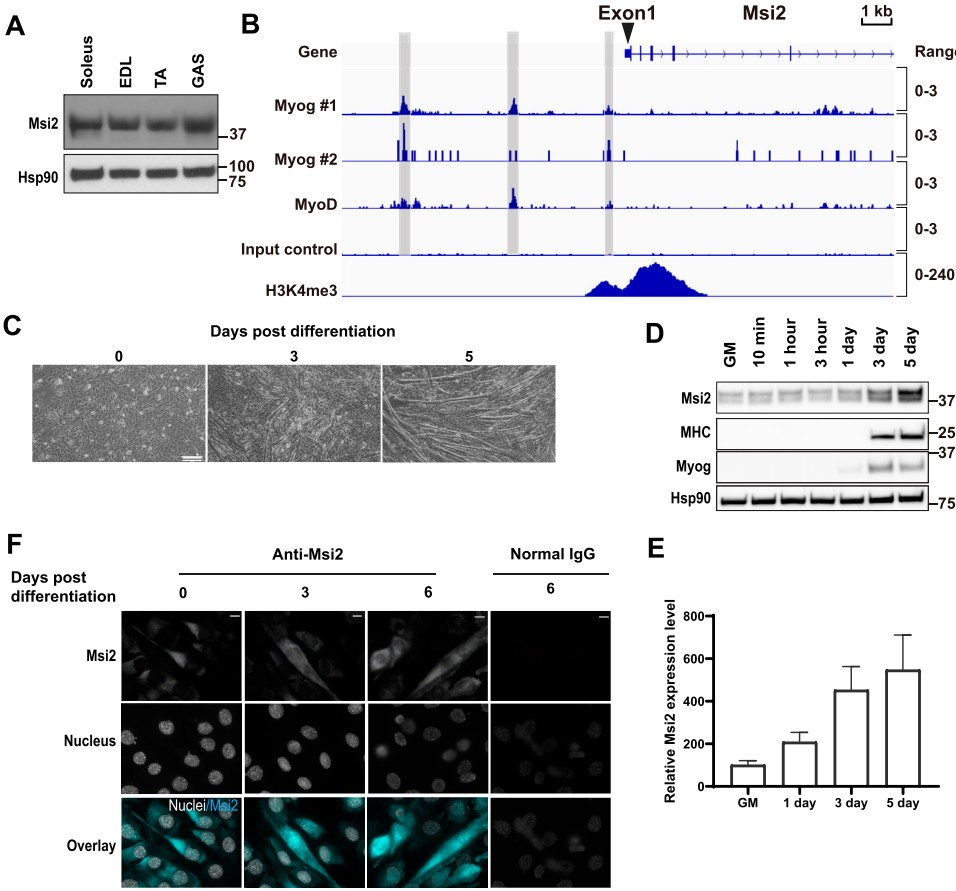

**Figure 1. Msi2 up-regulation during C2C12 differentiation.**
**(A)** Msi2 expression in skeletal muscles of mouse limbs analyzed by Western blotting on tissue lysates from the soleus, tibialis anterior, and gastrocnemius, along with that of the olfactory bulb from the brain. **(B)** MyoD and myogenin (Myog) bind to the *Msi2* gene regulatory regions in differentiated C2C12 cells. Shown are ChIP-seq data of Myog, MyoD, and H3K4me3 at *Msi2* gene in differentiated C2C12 cells from the ReMap Atlas (for Myog and MyoD) and ENCODE (for H3K4me3). Gray boxes highlight the regions bound by both Myog and MyoD. **(C)** Time course of C2C12 differentiation. Cells started to form myotubes at Day 3 and became highly fused at Day 5 after differentiation induction. Phase-contrast microscope images are shown. Scale bar = 500 $\mu$m. **(D)** Msi2 expression was up-regulated during C2C12 differentiation. Protein samples were collected at the indicated time points after differentiation induction. Myosin heavy chain and myogenin expression show the extent of differentiation. Hsp90 was used as a loading control. GM denotes the cells maintained in the Growth Media. **(E)** Quantification of relative Msi2 expression levels during myoblast differentiation. Gene expression was normalized to that of Hsp90 loading control at each time point. Msi2 level in GM culture was set as 100. Results from three independent experiments were included. **(F)** Msi2 expression was increased in both nuclear and cytoplasmic areas during differentiation. IF images of Msi2 and DNA stainings were collected with a confocal microscope at the indicated differentiation stage. Normal IgG was used as a negative control. Scare bar = 10 $\mu$m.
Source data are available for this figure.

and non–small-cell lung cancer (Kudinov et al, 2016; Kang et al, 2017). Although Msi2 is highly expressed in normal skeletal muscles (this study), its role in myogenesis has not been addressed previously. In this context, we found that Msi2 is essential for the differentiation and fusion of myoblasts by promoting autophagy. Consistently, mice deficient for Msi2 show a decrease in skeletal muscle mass and perform poorly in exercise endurance. Our study identifies Msi2 as a novel regulator of myogenesis and autophagy and provides new insights into our understanding of muscle development directed by MRFs.

# Results

### Msi2 expression in myoblast differentiation

To examine the contribution of Msi2 in muscle development, we first analyzed its expression in mouse skeletal muscles. Western blot (WB) analysis showed that Msi2 is highly expressed in all three hindlimb muscles tested, soleus, tibialis anterior, and gastrocnemius (Fig 1A). Consistent with its high expression level in skeletal muscles, we found that two MRFs, Myog and MyoD1, bind to the *Msi2*

gene promoter region (Fig 1B). These data implied that *Msi2* is activated during myogenesis and myocyte differentiation. Next, we examined Msi2 expression during myocyte differentiation in the murine C2C12 myoblast cell line. In an FBS-based media (Growth Media or GM), C2C12 cells expand as myogenic progenitors known as myoblasts. The progenitor cells undergo myocyte differentiation after incubation in media with horse serum (Differentiation Media or DM), and within 5 d, these myocytes form multinucleated elongated myotubes (Fig 1C). We found that Msi2 is expressed at a low level in myoblasts cultured in GM and that the expression is up-regulated within 24 h after the differentiation induction by DM (Fig 1D and E). Immunofluorescence staining of the Msi2 protein confirmed its up-regulation upon induction of differentiation (Fig 1F). Interestingly, C2C12 cells exhibited heterogeneity in the levels of Msi2 protein, and cells with higher Msi2 expression showed more differentiated morphology with more nuclear Msi2 staining at Days 3 and 6 after differentiation induction. Up-regulation of Msi2 was concomitant with that of Myog and myosin heavy chain (MHC; Fig 1D). In contrast, Msi1, the other family member of the mammalian Msi RBPs, is not expressed or up-regulated during C2C12 differentiation and no promoter binding of Myog or MyoD1 was observed (Fig S1A and B), indicating Msi2-specific roles in myogenic differentiation.

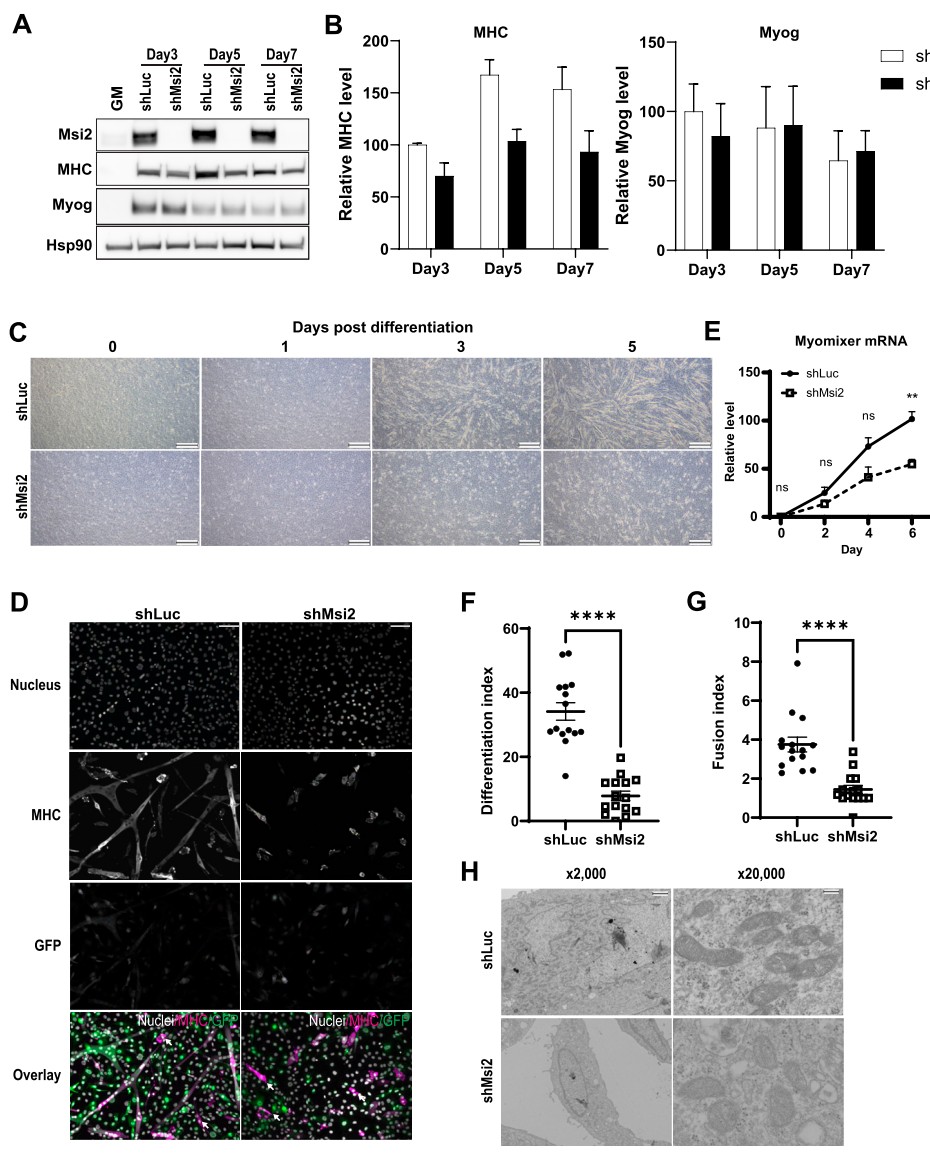

**Figure 2. Loss of Msi2 impairs C2C12 differentiation.**
**(A)** Msi2 knockdown (KD) reduced MHC protein expression but not Myog expression during myoblast differentiation. C2C12 cells lentivirally infected with a control shRNA (shLuc) or shRNA against Msi2 (shMsi2) were collected at different time points after differentiation induction and subjected to WB analysis. **(B)** Quantification of MHC and Myog expression levels normalized to those of Hsp90. Results from three independent experiments are shown. **(C)** Phase-contrast images of C2C12 cells infected with shLuc (top panels) or shMsi2 (bottom panels) at different time points after differentiation induction. Note that cells with Msi2 KD showed very little myotube formation. Scale bar = 500 μm. **(D)** Immunofluorescence staining of MHC showing reduced MHC-positive myocytes and myotube formation in the shMsi2 group compared with the shLuc control on Day 6 after differentiation induction. MHC staining shows differentiated myocytes and myotubes. GFP-positive cells are infected with shRNA lentivirus. DAPI was used to stain nuclei. Note that many cells with high MHC in the shMsi2 group were GFP-negative, that is, non-infected, cells (highlighted with white arrows). Scale bar = 100 μm. **(E)** Real-time PCR analysis of myocyte fusion gene *Myomixer* mRNA expression. At Day 6 of differentiation, the shMsi2 group showed significantly lower *Myomixer* expression. The expression levels were normalized to the expression levels of the beta-2 microglobulin gene at each time point. Data from three independent experiments are shown. **(F, G)** Reduced Differentiation (F) and Fusion (G) Indices in shMsi2-treated C2C12 cells at Day 6 after differentiation induction. Each dot represents data from one field shown in panel (D). Five fields were picked in each experiment, and data from three independent experiments are shown. An unpaired *t* test was performed between shLuc and shMsi2 groups. Scale bar = 100 μm. **(H)** Msi2 KD cells have less mature mitochondria. Representative electron micrographs of lentiviral shLuc- or shMsi2-treated C2C12 cells were taken on Day 5 after differentiation induction. Scale bar = 15 μm (×2,000) or 1.5 μm (×20,000). Source data are available for this figure.

## Msi2 is essential for the terminal differentiation of myocytes independent of MRF expression

Next, we investigated whether Msi2 is functionally required for myocyte differentiation using a lentiviral shRNA-mediated gene knockdown approach. 3 d after viral transfer of an shRNA against mouse Msi2 (shMsi2), the cells showed no detectable Msi2 protein expression, with knockdown (KD) continued up to 7 d (Fig 2A, Msi2). In Growth Media, shMsi2-expressing cells proliferate as much as those with control shRNA (shLuc), indicating that Msi2 is not essential for cell growth or survival of myoblasts (Fig S2A). After differentiation induction by switching from GM to DM, control shRNA-treated cells started to show elongated myotubes at as early as 3 d post-differentiation induction (Fig 2C, shLuc). The cells expressing shMsi2 also generated elongated cells, but they

appeared shorter in length and their numbers were much less than those with shLuc even after 5 d of differentiation, indicating a significant impairment of myotube formation in the absence of Msi2 (Fig 2C, shMsi2). MHC protein was up-regulated in shMsi2-expressing cells, but the magnitude of expression induction was not as robust as that of shLuc cells (Fig 2A and B, MHC). In contrast, Myog expression was comparable between the control and the Msi2 knockdown cells, suggesting the lower MHC induction is not due to a defect in Myog expression (Fig 2A and B, Myog). Consistent with the immunoblots and immunostaining of MHC proteins (Fig 2A and D), mRNA analysis of mouse MHC genes *Myh1* and *Myh4* confirmed their attenuated expression by the Msi2 knockdown (Fig S2B). In contrast, *Myod1* and *Myog* mRNA expression levels were comparable in both groups, suggesting the decreased *Myh* gene expression by Msi2 knockdown is downstream of the transcriptional regulation by the

myogenic transcription factors Myod1 and Myog (Fig S2B). Moreover, the expression level of the myocyte fusion gene *Myomixer* was reduced by Msi2 knockdown at the late stage of differentiation (Fig 2E). In contrast, we observed no decrease in viabilities of the Msi2 knockdown cells, and the levels of the cleaved PARP and caspase-3 proteins were not affected by the Msi2 loss during the differentiation process, indicating the differentiation defect is not due to increased cell death (Fig S2C).

To address whether the differentiation is impaired at a specific stage during myotube formation, we monitored differentiation status using immunofluorescence microscopy (Fig 2D). Consistent with the phase-contrast images shown in Fig 2C, shLuc-treated control cells generated MHC-positive elongated myotubes. In contrast, cells treated for Msi2 knockdown cells failed to generate mature myocytes and most of the MHC-positive cells, if any, were shorter and smaller in size (Fig 2D). We observed some elongated cells, but interestingly, most of them were GFP-negative, indicating that the myocytes that matured were not expressing the shRNA against Msi2 (Fig 2D, arrows). Differentiation Indices, defined as the ratio of nucleus numbers in MHC-positive cells to the total nuclei of all GFP-positive cells in an analyzed field, were significantly lower in the shMsi2 group than in the control group (Fig 2F). In addition, the shMsi2 cells exhibited a significantly lower Fusion Index, defined as the number of myonuclei in one MHC$^+$ cell, indicating an impact of Msi2 knockdown on the myocyte fusion step (Fig 2G), which is consistent with the decreased *Myomixer* expression (Fig 2E). Furthermore, electron microscopic images of the knockdown cells show distinct changes in mitochondria (Fig 2H). The mitochondria in control cells appeared elongated and contained complex crista structures with higher electron densities that are typical for mature myotubes. In contrast, those in the shMsi2 cells were round in shape and exhibited simple and scarce crista structures, implying defective mitochondrial remodeling in the Msi2 knockdown cells. Collectively, these results clearly demonstrated an essential role of Msi2 in the terminal differentiation of myocytes.

## Msi2 expression rescues the differentiation defects conferred by Msi2 knockdown

To exclude the possibility of an off-target effect by the lentiviral shMsi2 and to further confirm the functional requirement of Msi2 in myocyte differentiation, we investigated whether Msi2 overexpression (Msi2 OE) can rescue the differentiation defect caused by the knockdown. The C2C12 cells expressing exogenous Msi2 with Msi2 knockdown (shMsi2 + Msi2 OE, Fig 3A-iv and B-iv) exhibited as many elongated MHC-positive myotubes as those in the control cells (shLuc + vector control, Fig 3A-i and B-i), whereas the cells with shMsi2 and an empty vector control showed smaller myotubes with low MHC expression (shMsi2 + vector control, Fig 3A-iii and B-iii). A few cells in this treatment group still showed MHC expression, but they were typically smaller in size or contained only one nucleus, suggesting no mature myocyte formation. The vector control cells with Msi2 knockdown exhibited Differentiation and Fusion Indices of 10% and 1.3, respectively, whereas the Msi2 OE with Msi2 knockdown increased these indices to 21% and 2.1 (Fig 3C). In addition, Msi2 OE also reverted the mitochondrial morphology changes observed in the Msi2 KD cells (Fig S3). Collectively, these

results exclude the possibility of an off-target gene knockdown and demonstrate that Msi2 is required for differentiation and fusion into mature myocytes during the terminal differentiation step of myotube formation.

## Enforced expression of Msi2 promotes myotube formation

During the above-mentioned experiment to assess on-target shRNA effects, we noticed that Msi2 overexpression alone increased the formation of MHC-positive, multinucleated elongated myotubes (shLuc + Msi2 OE, Fig 3A-ii and B-ii). In fact, Differentiation and Fusion Indices were significantly higher in cells expressing shLuc and Msi2 OE than in control cells with shLuc and vector. We repeated the Msi2 overexpression experiment without the co-introduction of any shRNA constructs and observed a similar enhancement of myocyte differentiation (Fig 3D). In addition, Msi2 overexpression increased overall mitochondrion numbers and promoted their fusion in both naïve and Msi2 knockdown cells (Figs 3E and S3). Immunoblot analysis showed that the MHC protein is further up-regulated by Msi2 OE compared with the vector control (Fig 3F, MHC), confirming the direct and functional impact of Msi2 on mature myocyte formation. Msi2 overexpression did not lead to an increase in Myog protein expression (Fig 3F, Myog), providing further evidence for a role of Msi2 downstream of the MRF Myog.

## The RRM1 RNA-binding motif and the C-terminal domain of Msi2 are required for the promotion of myocyte differentiation

Because the Msi protein contains several domains for its molecular function (Okano et al, 2002; Kawahara et al, 2008), we next performed a structure–function relationship analysis. We introduced a retroviral vector encoding WT or a mutant Msi2 into C2C12 cells and observed their effects on differentiation (Fig 4A). The RNA binding–deficient mutant contains three point mutations at codons for Phe64, Phe66, and Phe69 in the RRM1 mutating each to a leucine residue, which leads to a loss of RNA-binding activity (Hattori et al, 2017). The Msi2 [1–190] mutant encodes only the first 190 aa and thus lacks the C-terminus domain. The ΔRRM1 and ΔRRM2 variants lack the first (21–101 aa) or second (110–187 aa) RRM, respectively. All the above Msi2 constructs are N-terminally fused with FLAG- and HA-tags. WB analysis using an anti-FLAG antibody showed the successful overexpression of all the Msi2 mutant proteins in the C2C12 cells (Fig 4B), except for the ΔRRM2 mutant that showed lower expression compared with the other mutants, despite its high retroviral infection (Fig 4C, GFP). On Day 3 after induction of differentiation, WT and ΔRRM2 Msi2 OE samples exhibited more MHC-positive cells, indicating enhanced differentiation (Fig 4C). Msi2 [1–190] and ΔRRM1 showed similar numbers of MHC-positive myocytes compared with a vector control. Interestingly, the expression of the RNA binding–deficient mutant resulted in attenuated differentiation compared with the vector control. Differentiation Indices were also significantly increased in WT and ΔRRM2 Msi2 OE samples but not in the others compared with the vector control (Fig 4D, Day 3 of differentiation). Because myocyte fusion mainly happens later in differentiation, Fusion Indices were comparable among all groups (Fig 4E). MHC detection by immunofluorescence correlated with MHC protein expression by WB, which also showed higher MHC

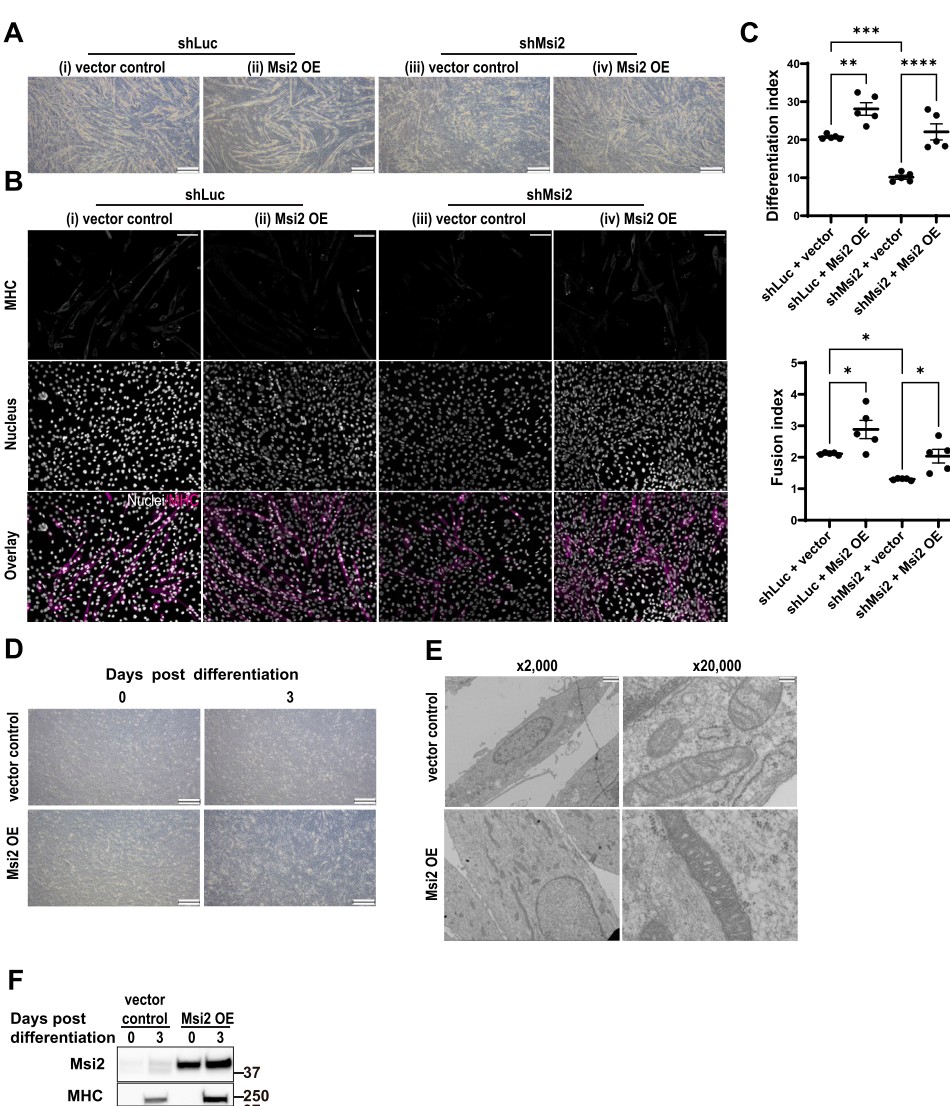

**Figure 3. Msi2 overexpression can suppress differentiation defects in Msi2 knockdown cells and promote C2C12 differentiation.**
**(A)** Phase-contrast images showing improved myotube formation by a retroviral Msi2 expression vector (Msi2 OE) in the shLuc- or shMsi2-treated C2C12 cells at Day 6 after differentiation induction. An empty retroviral vector was used as the control (vector control). Scale bar = 500 $\mu m$.
**(B)** Immunofluorescence staining of MHC protein in shLuc- or shMsi2-treated C2C12 cells infected with the empty control (vector control) or Msi2 expression vector (Msi2 OE) at Day 5 after differentiation induction. Differentiated myocytes and myotubes were stained positive for MHC. DAPI was used to stain nuclei. Scale bar = 100 $\mu m$.
**(C)** Differentiation and Fusion Indices in shLuc- and shMsi2-treated cells. Each dot represents the index from one field taken as shown in panel (B), and five independent fields were evaluated in each group. One-way ANOVA with Tukey's post hoc test was used. **(D)** Phase-contrast images showing augmented myotube formation by the retroviral Msi2 expression alone (Msi2 OE) at Day 3 after differentiation induction. A retroviral empty vector was used as the control (vector control). Scale bar = 500 $\mu m$.
**(E)** Representative electron micrographs of the cells with a retroviral control or an Msi2 expression vector (Msi2 OE) on Day 5 after differentiation induction. Scale bar = 15 $\mu m$ (×2,000) or 1.5 $\mu m$ (×20,000). **(F)** WB analysis showing Msi2 OE up-regulated MHC but not Myog expression on Day 3 after differentiation induction. Hsp90 was used as a loading control.
Source data are available for this figure.

expression levels in WT and ΔRRM2 than the others (Fig 4B, MHC). These data demonstrated that the RRM2 domain is dispensable for myoblast differentiation, whereas the RRM1 RNA-binding activity and C-terminus sequence of Msi2 are required for Msi2 to regulate myoblast differentiation.

## Msi2 regulates myoblast differentiation through autophagy

So far, our data indicated that Msi2 is not only essential but also sufficient for myoblast differentiation, without affecting the expression of MRFs. Because previous studies have shown that autophagy and mitophagy processes are required for myoblast differentiation (McMillan & Quadrilatero, 2014; Sin et al, 2016), we analyzed autophagosome formation by examining LC3A and LC3B expressions in Msi2 knockdown cells. LC3-II, the lipid-modified form of LC3, is an indicator of an intracellular autophagosome formation

level, which represents the extent of overall autophagy activity (Zois et al, 2011; Baeken et al, 2020; Klionsky et al, 2021). After differentiation induction, both non-lipidated LC3A-I and LC3A-II were elevated in control shLuc cells, whereas in Msi2 KD cells, LC3 protein levels were relatively unchanged (Fig 5A), suggesting attenuated initiation of autophagy by the loss of Msi2. The mRNA levels of LC3A were comparable between shLuc- and shMsi2-treated cells (Fig 5B), implying post-transcriptional regulation of LC3A protein levels by Msi2. LC3B mRNA levels were changed by Msi2 knockdown in the late (Day 5) but not the early (Days 1 and 2) differentiation stage. Immunofluorescence staining further confirmed that Msi2 knockdown cells showed weaker overall LC3A and LC3B staining intensities and significantly less cytoplasmic LC3 puncta per cell (Fig 5C and D), indicating attenuated autophagosome formation in the absence of Msi2. In addition, Msi2 OE increased both LC3A and LC3B autophagic fluxes in shLuc- and shMsi2-treated cells (Fig S4A and B),

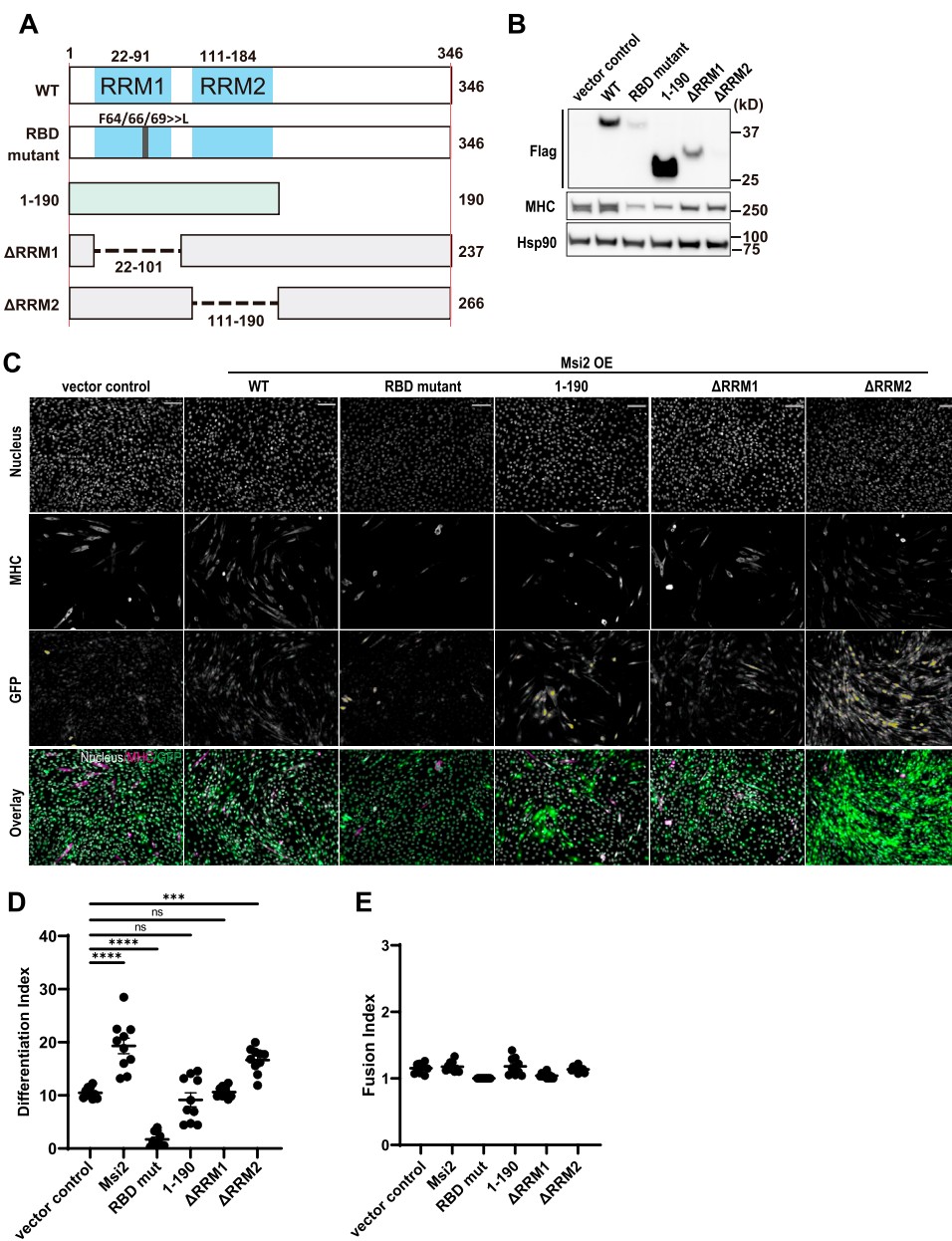

**Figure 4. Msi2 RRM2 RNA-binding domain is dispensable for the promotion of myogenesis.**
**(A)** Schematic illustration of the Msi2 mutants used. **(B)** Expression of WT and mutant Msi2 proteins. Samples were collected at Day 3 after differentiation induction. MHC expression levels show the extent of myoblast differentiation in each group. Hsp90 was used as a loading control. **(C)** Immunofluorescence staining of MHC in C2C12 expressing the indicated construct on Day 3 of differentiation. Note that vector control, WT, and ΔRRM2-expressing cells show many elongated MHC$^+$ myotubes. GFP-positive cells are infected cells with retrovirus for the indicated construct. DAPI was used to stain nuclei. Scale bar = 100 $\mu$m.
**(D, E)** Differentiation and Fusion Indices in cells expressing Msi2 mutant proteins. Expression of WT or ΔRRM2 Msi2 OE, but not the other Msi2 mutants, increased Differentiation Index (D) of C2C12 cells on Day 3 after differentiation induction. Fusion Indices (E) were comparable among all groups on Day 3 of differentiation. Each dot represents an index value from one field of cells as shown in Fig 4C, and the indices from five independent fields were used for analysis. One-way ANOVA and Tukey's multiple comparisons were performed. Source data are available for this figure.

demonstrating an autophagy-promoting function of Msi2. The expression level of a lysosomal membrane protein LAMP2 was comparable between control and Msi2 knockdown cells during differentiation, suggesting no overt defects in lysosome biogenesis in the absence of Msi2 (Fig S4C).

To examine whether the autophagy defect in Msi2 KD is the direct cause of the myoblast differentiation arrest, we activated autophagy and analyzed its impact on C2C12 differentiation. To induce autophagy, we used a small cell-permeable peptide Tat-Beclin 1-D11 (D11), which is known to activate autophagy by binding to the autophagy repressor GAPR1 to release Beclin 1 (Shoji-Kawata et al, 2013). A scrambled peptide Tat-Beclin 1-L11S (L11S) was used as a negative control. With the D11 peptide treatment, both control and Msi2 KD cells generated more and larger myotubes on Day 5 post-

differentiation induction (Fig 6A and B). Immunoblot analysis confirmed that LC3A-II and LC3B-II levels in control and Msi2 KD cells were increased by the D11 treatment, confirming activated autophagy (Fig 6C). Autophagy activation by the D11 peptide treatment significantly increased the Differentiation Index of Msi2 KD compared with the L11S scrambled control peptide from 13% to 29%, which is comparable to 34% in control cells (Fig 6D), suggesting the differentiation defect of Msi2 KD cells was fully rescued by autophagy induction. Tat-Beclin 1-D11 treatment also significantly increased the Fusion Index of Msi2 KD cells twofold, which is a partial rescue compared with 4.1 ± 0.43 of the control (Fig 6E). Surprisingly, Tat-Beclin 1-D11 alone significantly increased both Differentiation and Fusion Indices in the control cells (Fig 6D and E), implying a differentiation-promoting function of autophagy. Taken

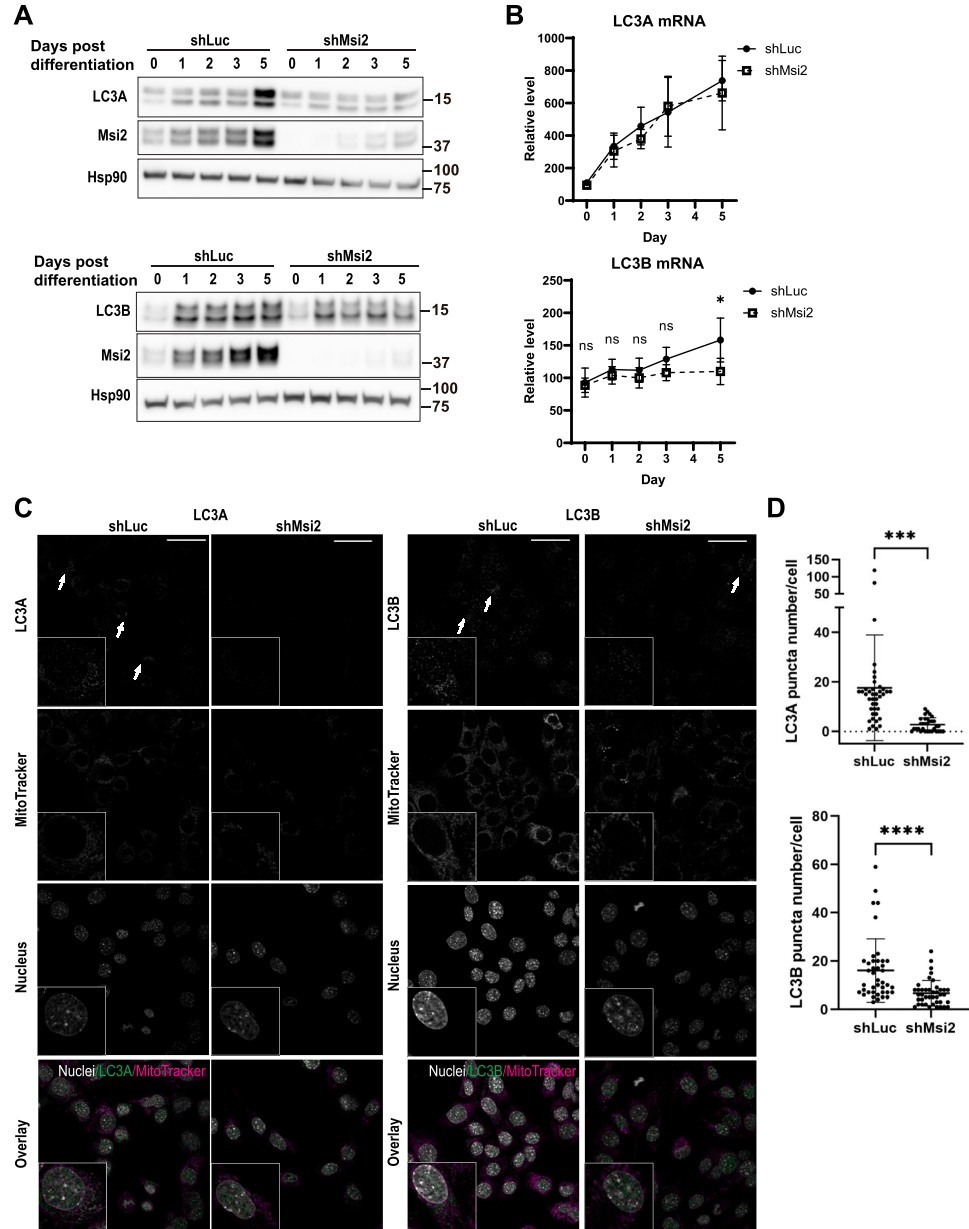

**Figure 5. Msi2 regulates autophagy during C2C12 differentiation.**
**(A)** WB analysis of LC3 proteins during myoblast differentiation. Both LC3A (top panel) and LC3B (bottom panel) proteins were up-regulated upon differentiation induction. Msi2 knockdown attenuated LC3 up-regulation during differentiation. **(B)** Real-time PCR analysis showing LC3 mRNA levels. Msi2 KD did not affect their gene expression at earlier phases of the myoblast differentiation. Data from three independent experiments are shown. **(C)** Immunofluorescence staining of LC3A in control and Msi2 KD C2C12 at Day 1 of differentiation. White arrows indicate LC3A puncta in the cytoplasmic area. Scare bar = 40 μm. **(D)** Msi2 KD reduces cytoplasmic LC3A and LC3B punctate staining. Each dot represents the punctum number in one cell. Cells from three or five different fields in groups with LC3A or LC3B staining were analyzed, respectively. For LC3A, n = 40 and n = 32 for shLuc and shMsi2, respectively; for LC3B, n = 41 each for shLuc and shMsi2.
Source data are available for this figure.

together, the Tat-Beclin 1-D11 peptide successfully rescued both autophagy and differentiation in Msi2 KD cells, demonstrating that Msi2 regulates myoblast differentiation via autophagy through the up-regulation of LC3 protein expression.

### Reduced endurance capacity in Msi2 gene-trap mutant mice

To investigate the role of Msi2 in skeletal muscle in vivo, we used an *Msi2* mutant mouse model generated via a gene-trap mutagenesis strategy (Ito et al, 2010). From heterozygous intercrosses, we obtained 408 WT, 700 heterozygous mutant, and 87 homozygous mutant (KO) mice in total at weaning, indicating a skew in genotype distribution because of decreased survival before weaning (chi-square test, $P < 0.0001$). Of mice that survived until weaning, KO mice

were underweight, and exhibited kyphosis and reduced mobility, implying a defect in muscular function. Consistently, soleus muscles from KO mice were significantly smaller and presented a pale color compared with WT littermate controls (Fig 7A and B). Interestingly, we did not see a size difference in extensor digitorum longus (EDL) muscles, implying predominately slow-twitch muscles, but not fast-twitch, are affected by Msi2 deficiency. WB analysis confirmed that Msi2 expression was lost in the skeletal muscle tissues in KO mice, whereas Msi1 expression was comparable (Fig S5A). Next, we performed several locomotor activity assays. In an open-field activity paradigm, the total travel distance for KO mice was 7.64 m ± 0.52 m, whereas it was 20.63 m ± 0.81 m for the littermate controls for the duration of 30 min (Fig S5B). Vertical activity counts, which are relevant to hindlimb strength, are significantly

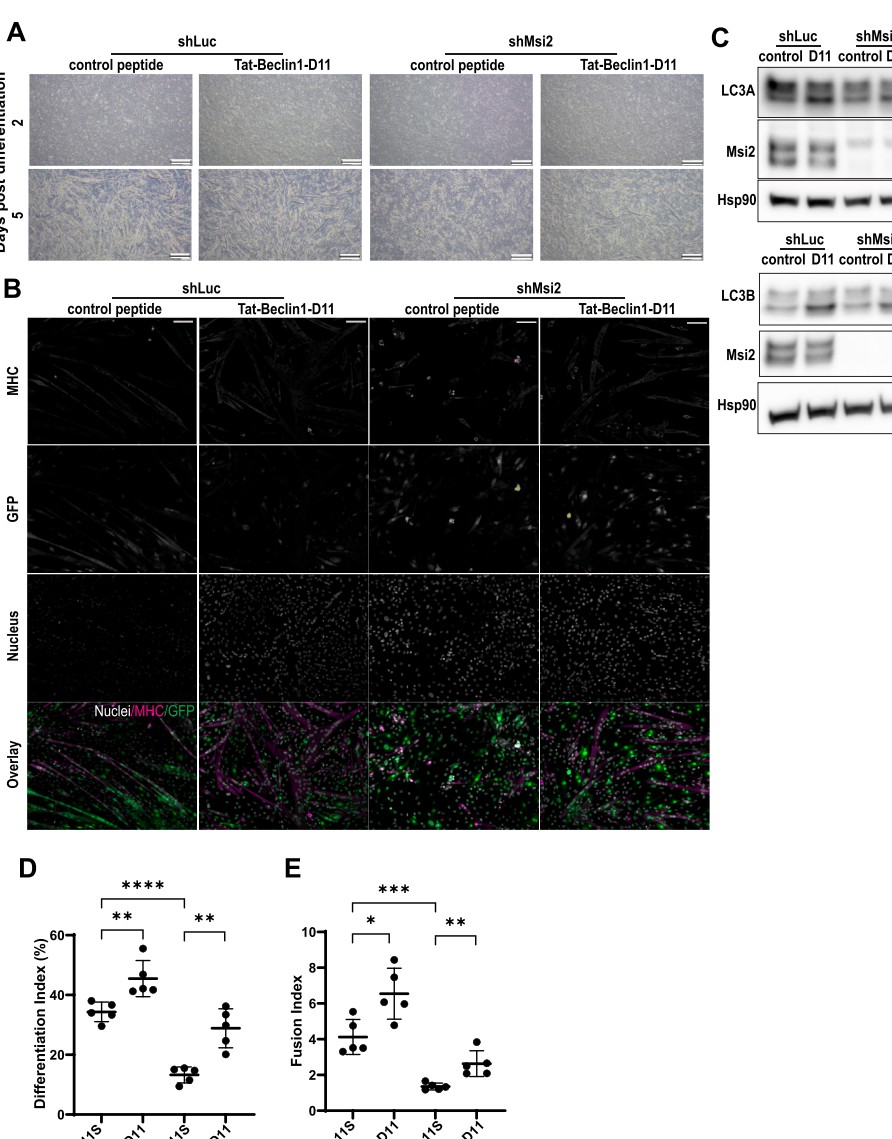

Figure 6. Activation of autophagy suppressed the differentiation defects by Msi2 knockdown.
**(A)** Myotube formation defect in Msi2 KD C2C12 cells is rescued by autophagy activation. Shown are bright-field images of shLuc or shMsi2 C2C12 treated with a scrambled control peptide (Tat-Beclin 1-L11S) or Tat-Beclin 1-D11 peptide at Days 2 and 5 after differentiation induction. Scare bar = 500 μm. **(B)** Immunofluorescence staining of MHC at Day 5 after differentiation induction in cells treated with a Tat-Beclin 1-D11 peptide. Note that the Tat-Beclin 1-D11 peptide–treated Msi2 KD cells had as many elongated MHC⁺ myotubes as the naïve shLuc control. GFP-positive cells are virus-infected cells expressing shLuc or shMsi2. DAPI was used to stain nuclei. Scale bar = 100 μm. **(C)** WB analysis of LC3 proteins in cells treated with a Tat-Beclin 1-D11 peptide. The treatment increased LC3A-II or LC3B-II (bottom bands) levels in both shLuc and shMsi2 KD cells at Day 2 after differentiation induction as shown in Fig 6A. **(D, E)** Differentiation (D) and Fusion (E) Indices in cells treated with a Tat-Beclin 1-D11 peptide at Day 5 of differentiation. Each dot represents an index value from one field of cells shown in Fig 6B, and the indices from five independent fields were used for analysis.
Source data are available for this figure.

lower in KO mice compared with WT in both males and females (Fig S5C). In contrast, mice exhibit no overt changes in stereotypic behaviors regardless of their genotypes and sexes, suggesting that lower locomotor activity is due to physical, not neurological, defects in the mutants (Fig S5D). To further analyze the locomotor defect, we tested 16-wk-old mice with a downhill exhaustion treadmill run (Fig 7C). The littermate control mice could run for more than 100 m, whereas *Msi2* mutant mice met the criteria for exhaustion by 33 m ± 5.21 m regardless of their sexes, indicating that the mutants had reduced capacity for movement, consistent with defects in skeletal muscle and myocyte differentiation from Msi2 deficiency shown above.

To further investigate the impact of Msi2 loss on muscle functions, we performed histological analysis of the soleus, as well as tibialis anterior (TA) and EDL muscles (Figs 7D and S5E). Total cross-sectional area, muscle fiber number, and average fiber area were all smaller in the Msi2 KO muscles compared with WT, indicating muscular atrophy phenotypes (Fig S5F and G). In addition to the smaller muscle fiber sizes in the KO mice, we also examined whether Msi2 KO mice have a change in fiber-type distribution. We performed immunofluorescence staining of type I fibers (slow, oxidative), type IIA fibers (fast, oxidative and glycolytic mixed), and type IIB fibers (fast, glycolytic) using specific antibodies to each MHC subtype (Fig 7E). Type IIX fibers (fast, glycolytic) appear unstained. Surprisingly, we observed a significant increase in type I fibers at the expense of type IIA fibers in the soleus (Fig S5H). In soleus muscles, type I, type IIA, and type IIX fibers were 43%, 34%, and 23% in KO, whereas those in the WT soleus were 34%, 55%, and 10%, respectively (Fig 7F, top panel). A similar change was observed in fast muscles TA and EDL as well; in WT, we observed no detectable

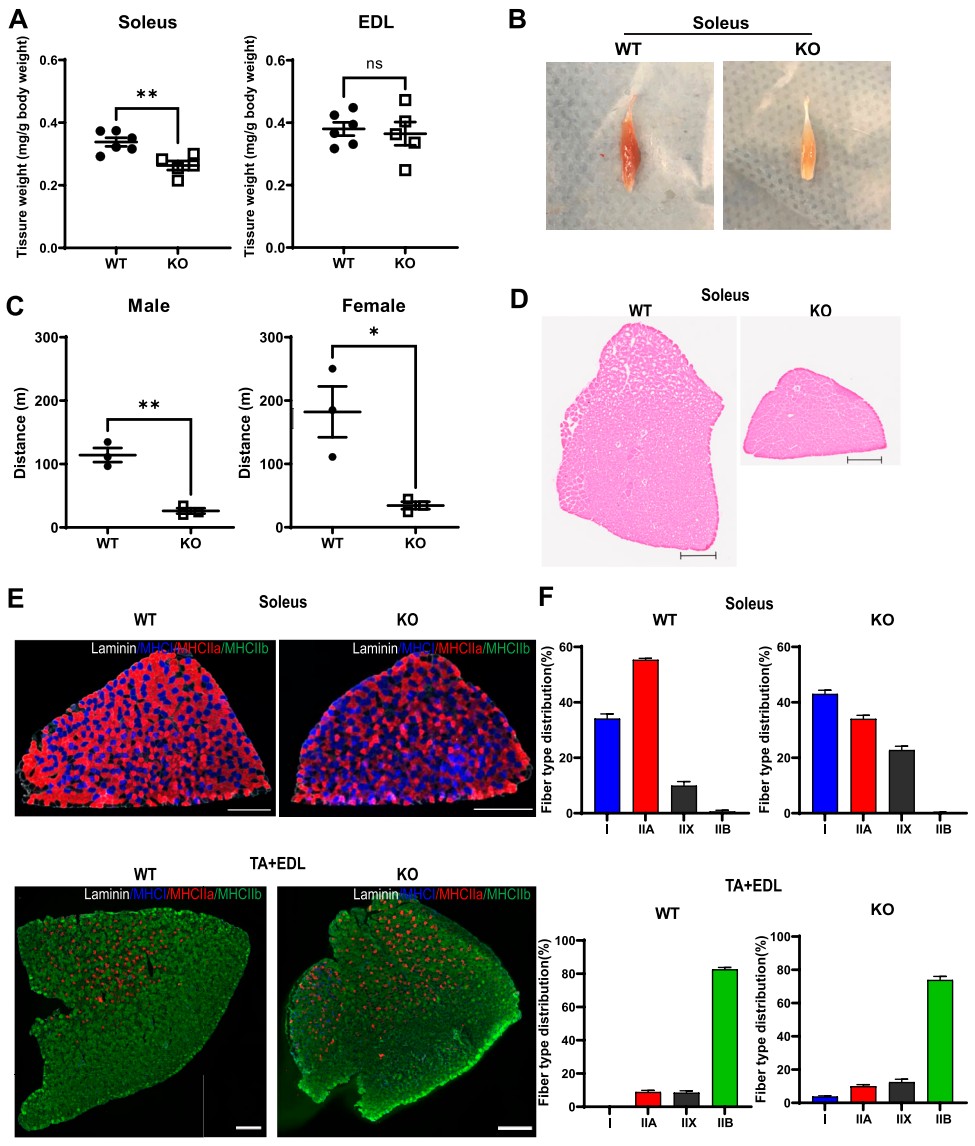

**Figure 7. Loss of Msi2 leads to smaller soleus and decreased exercise capacity in mice.**
**(A)** Weights of soleus and extensor digitorum longus muscles from WT and Msi2 KO mice, age 4–9 mo. Muscle weights were normalized to the body weight of each mouse. For WT and KO, n = 6 and n = 5, respectively. **(B)** Soleus from WT and Msi2 KO mice. Note the pallor appearance of the KO soleus. **(C)** Running distance in a treadmill fatigue assay. Msi2 KO mice had less endurance than WT mice. Each dot represents a run time from an individual animal. N = 3 for each group, at 4 mo of age. **(D)** Representative images of hematoxylin-and-eosin staining of WT and Msi2 KO mouse soleus muscle sections. Scale bar = 200 $\mu m$. **(E)** Representative immunostaining images of soleus, and TA and EDL muscle cross-sections from WT and Msi2 KO mice. Tissues were immunostained with laminin (white), MHC I (type I, blue), MHC IIa (type IIA, red), and MHC IIb (type IIB, green). Type IIX fibers appear unstained. Scale bar = 200 $\mu m$ (soleus) or 400 $\mu m$ (TA + EDL). **(F)** Fiber-type distribution in WT and Msi2 KO muscles. Two individual mice were included in each group. Muscles were dissected from two hindlimbs of two mice, and four individual sections were stained for MHC expression for fiber typing.

type I fibers, but nearly 5% of the fibers were type I in TA and EDL muscles from KO mice. The proportion of type IIB fibers also significantly decreased from 83% in WT to 74% in KO (Fig 7F, bottom panel). These fiber typing results indicate a shift in muscular cell fates from fast-twitch to slow-twitch fibers in the Msi2 KO skeletal muscles. Collectively, our results clearly demonstrated that Msi2 is essential for mature myocyte formation and thus indispensable for robust muscular activities in vivo.

## Discussion

In the present study, we investigated the role of the RBP Msi2 in myoblast differentiation and skeletal muscle functions. We found that *Msi2* is a potential target gene for the myogenic regulatory transcription factor Myog. Consistent with this finding, Msi2 knockdown in C2C12 severely impaired the terminal differentiation

into mature myocytes, and genetic loss of Msi2 led to apparent physiologic defects in skeletal muscle function in vivo. Interestingly, we observed a shift from type II fast-twitch to type I slow-twitch fibers in Msi2 KO muscle tissues. Type I slow-twitch fibers are often associated with muscular contractile endurance because they generate lower contractile forces (ATP demand) and have greater oxidative capacity (ATP supply). It is notable, perhaps counterintuitive, that Msi2 KO mice exhibit significantly diminished running capacity and cage activities. However, limited running capacity and cage activities in the Msi2 KO mice likely have more to do with smaller muscle size, fewer fiber numbers, and small fiber cross-sectional area. Another possibility is that the MHC I–positive type I fibers in Msi2 KO do not have high-quality mitochondria to meet energetic demands during activity because of mitochondrial defects.

During myoblast differentiation, Msi2 overexpression not only alleviated the defects caused by the Msi2 knockdown but also

promoted terminal differentiation of C2C12 cells. At the molecular level, either Msi2 knockdown or overexpression did not affect the protein expressions of the canonical myogenic transcription regulator MRFs, including Myog. Instead, we found that the Msi2 loss led to the decreased expressions of LC3 proteins, essential components for autophagosome formation, and that forced activation of autophagy suppressed the differentiation defects in the Msi2 knockdown cells. These results clearly demonstrated an indispensable role of Msi2 as a novel post-transcriptional regulator of autophagy in myoblast differentiation, and thus provided the first definitive genetic evidence for the essential functions of this RBP in muscle development and function.

The Msi family of proteins has two RRMs in the N-terminal region, which are functionally essential for RNA binding in vitro (Kawahara et al, 2008; Ohyama et al, 2012). The RRM2 domain has much weaker RNA-binding activity compared with that of RRM1 and thus is considered to primarily assist the RNA binding by the RRM1 (Nagata et al, 1999; Zearfoss et al, 2014). These results are consistent with our finding that the RRM2 is dispensable for promoting C2C12 differentiation (Fig 4C and D). In mammalian species, the C-terminal region of Msi2 is only 56% identical to that of Msi1 and its functions are not fully defined (Sakakibara et al, 2001). Our data showed the overexpression of an Msi2 mutant lacking the C-terminal region inhibited C2C12 differentiation (Fig 4C and D), providing new insight into the function of the Msi2 C-terminal domain. In Msi1, the C-terminal region contains a domain required for the interaction with the Poly(A)-binding protein, and interestingly, this region is not conserved in Msi2 (Kawahara et al, 2008; Cragle et al, 2019). It remains to be elucidated how the C-terminal region of Msi2 mediates differentiation-promoting function in myoblasts.

It is well established that RBPs have essential functions in the post-transcriptional regulation of myogenesis (Apponi et al, 2011; Shi & Grifone, 2021). For instance, ELAVL1/HuR and AUF1 regulate myogenesis by modulating the expressions of key factors such as MRFs and MEF2 (Figueroa et al, 2003; Panda et al, 2014). It is poorly defined how autophagy activation during myogenesis can be regulated by RBPs. To date, very few studies have demonstrated functional contributions of RBPs to autophagy in muscle development or myogenic differentiation, whereas several groups have reported RBP functions in autophagy. In human hepatocellular carcinoma cells, ELAVL1 regulates both autophagosome formation and autophagic flux via mediating translation of the autophagy-related genes (ATGs) ATG5 and ATG12 (Ji et al, 2019). ELAVL1 inhibits the maturation process of microRNA (miR)-7, which suppresses autophagy induction via the AMPK-mTOR pathway and directly binds to the 3'-UTRs of ATG4A, ATG7, and ULK2 mRNAs (Choudhury et al, 2013; Gu et al, 2017). Because Elavl1 and Msi2 proteins physically interact and functionally cooperate, it is possible that Msi2 coordinates autophagy in myoblasts by modulating miR-7 expression and function.

We also showed that ATG8/MAPL1LC3A and LC3B protein expressions are regulated by Msi2, suggesting a new functional link of RBP regulation of autophagy in myogenesis. The lipidation of LC3 protein has been extensively studied for its importance in the initiation of autophagy. At the level of gene expression, the transcription factor FoxO3 has been shown to play a pivotal role in skeletal muscle under AMPK-mTOR signaling (Mammucari et al,

2007; Sanchez et al, 2012). However, other modes of LC3 regulation, including those at the post-transcriptional steps, are poorly defined. One study showed that miR-204 can regulate LC3B expression through a direct binding to LC3B mRNA in kidney cancer (Mikhaylova et al, 2012). In the present study, we showed that Msi2 knockdown decreased LC3 protein without affecting its mRNA level at the early differentiation stage, suggesting post-transcriptional regulation of LC3 expression. Future studies should focus on the molecular mechanism of how Msi2 controls LC3 expression.

Mammalian species have multiple isoforms of LC3 proteins encoded by paralogous genes. In the human genome, three *LC3* genes have been identified, whereas there are only two genes in rodents (*LC3A* and *B*). Among these isoforms, the molecular mechanism of LC3B in autophagy has been extensively studied and widely used for monitoring autophagic flux (Klionsky et al, 2021). Although LC3A and LC3B proteins share similar primary structures, the tissue expression patterns of LC3A and LC3B are distinct (Zois et al, 2011). Previous studies have demonstrated different roles of LC3A and LC3B in autophagy, and it is therefore proposed that LC3 functions should be addressed separately and carefully in autophagy research (Weidberg et al, 2010; Schaaf et al, 2016). For example, LC3A and LC3B show distinct subcellular distribution and little colocalization in many cancer cell lines, as well as fibroblasts and umbilical vein endothelial cells (Koukourakis et al, 2015). In the present study, we found both LC3A and LC3B proteins were upregulated during myoblast differentiation but with different magnitudes; during C2C12 differentiation, *LC3A* mRNA increased sixfold, whereas *LC3B* increased by only 50%. These data imply that LC3A is the major autophagosomal protein in myogenesis. Nonetheless, Msi2 loss results in the reduced expression of both LC3A and LC3B proteins, which negatively regulate myoblast differentiation.

Much evidence has established that autophagy is essential for skeletal muscle development (Masiero & Sandri, 2010; McMillan & Quadrilatero, 2014). It has not been fully defined, however, whether autophagy activation will impact myogenesis. One study showed that autophagy activation by Atg5 overexpression improves motor function in mice (Pyo et al, 2013), raising the possibility that its activation can promote myogenesis. On the contrary, several other studies proposed that excessive autophagy pose negative impacts on skeletal muscle; in muscular atrophy and cachexia, autophagic activation causes loss of muscle mass (Sandri, 2010; Penna et al, 2013). In this context, our study demonstrated that direct autophagy activation using a Tat-Beclin 1 peptide promoted myoblast differentiation and Msi2 OE increased LC3 fluxes and helped the generation of mature myocytes. Our findings may also indicate context-dependent roles of autophagy activation in normal myoblast differentiation, unlike those in pathological conditions such as cachexia.

In summary, our study identified Msi2 as a novel regulator of myogenesis. This factor coordinates autophagy activation to promote myogenic differentiation and cell fusion of myocytes. Loss of Msi2 results in muscular atrophy and fiber-type switching to type I fibers. Because this is the first report on the role of Msi2 in autophagy regulation in myogenesis, these findings could extend to Msi2's function in cell fate decisions of other cell types including neural, hematopoietic, and leukemic stem cells regarding autophagy. This work also provided new insights into the expression

regulation of autophagy proteins LC3A and LC3B during myoblast differentiation and may open a new avenue for a therapeutic strategy in treating myopathies accompanied by abnormal autophagy.

# Materials and Methods

### Cell culture

The mouse C2C12 myoblast cell line was obtained from the ATCC (CRL-1772) and maintained in Growth Media (GM; DMEM with high glucose, 15% FBS [VWR], 1% non-essential amino acids, 100 IU/ml penicillin, and 100 $\mu$g/ml streptomycin) in a humidified incubator with 5% $CO_2$ at 37°C. For cell proliferation assays, the cells were plated in 96-well plates at a density of $1 \times 10^4$ cells/ml in GM. Cells were trypsinized and counted at the indicated time points. Media were changed every 3 d. To induce differentiation of C2C12 myoblasts into myotubes, the cells were plated in a six-well plate at $4 \times 10^4$ cells/ml in GM. After 24 h, the media were changed to Differentiation Media (DM; DMEM with 2% horse serum [HyClone], 1% nonessential amino acids, and 100 IU/ml penicillin and 100 $\mu$g/ml streptomycin). DM was replenished every 3 d during the differentiation period. For autophagy induction during C2C12 differentiation, C2C12 cells were plated in a six-well plate and differentiated as described above. At 24 h after the induction, cells were treated with 20 $\mu$M Tat-Beclin 1-L11S or Tat-Beclin 1-D11 for 5 h, followed by fresh DM replenishment as described above. All medium components were purchased from Corning or Nacalai Tesque unless otherwise described. For autophagy flux analysis, cells were treated with 100 nM bafilomycin or DMSO as a negative control for 3 h before harvesting for protein extracts.

### Viral constructs and production

Retroviral MSCV-IRES-EGFP and lentiviral FG12-UbiC-GFP vectors were obtained from Addgene (plasmid numbers #20762 and #14884, respectively). Mouse Msi2 cDNA was cloned into the MSCV vector, and lentiviral shRNA constructs were cloned into FG12 essentially as described previously (Ito et al, 2010). The target sequences are 5′-AGTTAGATTCCAAGACGA-3′ for mouse *Msi2* (shMsi2) and 5′-CTGTGCCAGAGTCCTTCGATAG-3′ for luciferase as a negative control (shLuc). Viruses were produced in 293FT cells transfected using polyethylenimine with viral constructs along with VSV-G, Gag-Pol, and Rev (in the cases of FG12) constructs as described before (Ito et al, 2010).

For viral transduction, cells were plated at a density of $2 \times 10^4$ cells/ml in a 60-mm dish and cultured for 24 h before infection. At 48 h post-infection, cells were monitored for GFP fluorescence using a microscope or flow cytometry. Then, cells were collected and replated for further analysis.

### Antibodies

The following antibodies were used: rabbit monoclonal anti-MSI2 (EP1305Y; Abcam); rat monoclonal anti-Msi1 (14989682; Thermo Fisher Scientific); mouse monoclonal anti-Hsp90 (F-8; Santa Cruz Biotech); rabbit monoclonal anti-LC3A (D50G8; Cell Signaling Technology); rabbit monoclonal anti-LC3B (43566; CST); rabbit polyclonal anti-PARP (9542; Cell Signaling Technology); rabbit monoclonal anti-caspase-3 (14220; Cell Signaling Technology); rat monoclonal anti-Lamp2 (ABL-93; Developmental Studies Hybridoma Bank); mouse monoclonal anti-MHC (MF20; Developmental Studies Hybridoma Bank); mouse monoclonal anti-MHC1 IgG2 (BA-F8; Developmental Studies Hybridoma Bank); mouse monoclonal anti-MHC2A IgG1 (SC-71; Developmental Studies Hybridoma Bank); mouse monoclonal anti-MHC2B IgM (BF-F3; Developmental Studies Hybridoma Bank); mouse monoclonal anti-Myog (F5D; Developmental Studies Hybridoma Bank); mouse monoclonal anti-FLAG (F1804; Sigma-Aldrich); goat anti-mouse IgG, HRP-conjugated (31342; Thermo Fisher Scientific); goat anti-rabbit IgG, HRP-conjugated (A16140; Thermo Fisher Scientific); goat anti-rabbit IgG, Alexa Fluor 594–conjugated (A11037; Thermo Fisher Scientific); goat anti-mouse IgG, Alexa Fluor 594–conjugated (A11032; Thermo Fisher Scientific); goat anti-mouse IgG1, Alexa Fluor 568–conjugated (A21124; Thermo Fisher Scientific); goat anti-mouse IgG2B, Alexa Fluor 647–conjugated (A21242; Thermo Fisher Scientific); goat anti-mouse IgM, Alexa Fluor 488–conjugated (A21042; Thermo Fisher Scientific); normal mouse IgG (I8765; Sigma-Aldrich); normal rabbit IgG (P120-101; Bethyl Laboratories).

### Western blotting

Cell lysates were prepared with NP-40 lysis buffer (50 mM Tris–HCl, pH 7.4, 150 mM NaCl, 5 mM EDTA, and 1% NP-40) with protease and phosphatase inhibitors (Nacalai Tesque). The BCA assay was used to determine protein concentrations (FUJIFILM Wako Pure Chemicals). After separation on SDS–PAGE, proteins were transferred to a polyvinylidene difluoride membrane using a semi-dry transfer system (iBlot2; Thermo Fisher Scientific) for immunoblotting. A chemiluminescence signal was detected using EzWestLumi Plus (2332638; ATTO) with ChemiDoc MP Imaging System (Bio-Rad).

### Immunofluorescence staining

For immunofluorescence staining, C2C12 cells were plated in a six-well plate with a glass coverslip at the bottom of each well before differentiation induction as described above. After 5 or 6 d in DM, cells were fixed in 4% PFA in PBS for 15 min at RT, washed three times with PBS, and permeabilized in PBS with 0.5% Triton X-100 for 15 min at RT. Samples were then blocked using Blocking One (Nacalai Tesque) for 30 min, and stained at RT for 1 h with primary antibody followed by incubation with an Alexa Fluor–conjugated secondary antibody and DAPI (Invitrogen). Samples were washed three times with PBS between primary and secondary antibody incubations. Coverslips were mounted onto slides using anti-fade mounting media containing 0.2% n-propyl gallate (FUJIFILM Wako Pure Chemicals). Fluorescence images were captured on AxioImager Z1 with Apotome.2 (Zeiss). For mitochondrion staining, cells were incubated with 200 nM MitoTracker Deep Red (Thermo Fisher Scientific) in a humidified incubator with 5% $CO_2$ at 37°C for 25 min before the fixation step as described above. For Differentiation and Fusion Indices, images were analyzed using ImageJ software,

version 1.53q. Nucleus and cell counting were performed only in GFP-positive, that is, virally transduced, cells. Quantifications of LC3 puncta were performed using ImageJ with a published macro (Dagda et al, 2008; Chu et al, 2009). For the muscle fiber typing experiment, muscles were isolated, embedded in OCT compound followed by snap freezing in isopentane under liquid nitrogen cooling, and cryosectioned. The staining procedure was the same as for C2C12 cells except for the fixation step. The primary antibodies targeting MHC I, MHC IIa, and MHC IIb are different immunoglobulin subtypes such as IgG2, IgG1, and IgM, respectively. Different Alexa Fluor–conjugated secondary antibodies targeting specific immunoglobulin subtypes were used to costain MHC I, IIa, and IIb on the same section.

### Electron microscopy

C2C12 cells were plated in a 100-mm dish at a density of $4 \times 10^4$ cells/ml in GM. After 24 h, the media were changed to DM to induce differentiation. After 5 d of differentiation, the cells were fixed with 2.5% glutaraldehyde (TAAB Laboratories Equipment Ltd) and postfixed with 1% osmium tetroxide (TAAB Laboratories Equipment Ltd) at 4°C. After en bloc staining with 1% uranyl acetate, the cells were dehydrated with a series of ethanol gradients, followed by propylene oxide, and embedded in Epon 812 resin (TAAB Laboratories Equipment Ltd). The ultrathin sections were stained with 2% uranyl acetate and lead citrate, and observed using Hitachi HT-7700 at 80 kV.

### ChIP-seq data analysis

ChIP-seq data for Myog and MyoD binding sites in differentiated C2C12 cells were retrieved from ReMap Atlas (dataset GSE44824) (Hammal et al, 2022). The H3K4me3 ChIP-seq data of differentiated C2C12 cells were obtained from ENCODE (datasets ENCSR000AHT) (He et al, 2020). Figures were generated using Integrated Genomics Viewer, version 2.14.0 (Robinson et al, 2011).

### Real-time and standard RT–PCR analysis

Total cellular RNAs were purified using a TRI Reagent (Sigma-Aldrich), and cDNAs were prepared from equal amounts of RNAs using Superscript IV Reverse Transcriptase (Thermo Fisher Scientific). Quantitative real-time PCRs were performed using EvaGreen qPCR Master Mix (Bio-Rad) on StepOnePlus (Thermo Fisher Scientific). Results were normalized to the level of $\beta$-2 microglobulin. PCR primer sequences are as follows. All primer pairs gave a specific PCR product with a single peak in melting curve analysis.

mB2m-F, 5′-ACCGGCCTGTATGCTATCCAGAA-3′
mB2m-R, 5′-AATGTGAGGCGGGTGGAACTGT-3′
mMyog-F, 5′-CTAAAGTGGAGATCCTGCGCAGC-3′
mMyog-R, 5′-GCAACAGACATATCCTCCACCGTG-3′
mMyh1-F, 5′-ATGAACAGAAGCGCAACGTG-3′
mMyh1-R, 5′-AGGCCTTGACCTTTGATTGC-3′
mMyh3-F, 5′-TGAACAGATTGCCGAGAACG-3′
mMyh3-R, 5′-GGAGAATCTTGGCTTCTTCGTG-3′
mMyod1-F, 5′-TTCTTCACCACACCTCTGACA-3′
mMyod1-R, 5′-GCCGTGAGAGTCGTCTTAACTT-3′

mMyh4-F, 5′-CACCTGGAGCGGATGAAGAAGAAC-3′
mMyh4-R, 5′-GTCCTGCAGCCTCAGCACGTT-3′
mMap1lc3a-F, 5′-GACCGCTGTAAGGAGGTGC-3′
mMap1lc3a-R, 5′-CTTGACCAACTCGCTCATGTTA-3′
mMap1lc3b-F, 5′-TTATAGAGCGATACAAGGGGGAG-3′
mMap1lc3b-R, 5′-CGCCGTCTGATTATCTTGATGAG-3′
mMyomixer-F, 5′-GTTAGAACTGGTGAGCAGGAG-3′
mMyomixer-R, 5′-CCATCGGGAGCAATGGAA-3′

### Mice

The Msi2 mutant mice, B6;CB-*Msi2*$^{Gt(pU-21T)2Imeg}$, established via gene-trap mutagenesis, were bred, and genotyped as previously described (Ito et al, 2010), and maintained on a 12-h:12-h light:dark cycle in the facilities of the University Research Animal Resources at University of Georgia, and of the Institute for Life and Medical Sciences at Kyoto University. All mice were 5–20 wk old, age-matched, and randomly chosen for experimental use. No statistical methods were used for sample size estimates. All animal experiments were performed according to protocols approved by the University of Georgia and the Kyoto University Institutional Animal Care and Use Committees.

### Open-field activity test

Open-field activity monitors (16 × 16 inch SuperFlex Open Field; Omnitech Electronics) with infrared beams to track horizontal and vertical mouse movements were used for the experiments. The same handler performed all experiments. Up to four mice of the same sex were placed in separate, adjacent testing chambers, initiating a 30-min test session: horizontal travel distance, horizontal activity count (horizontal beam breaks), ambulatory activity count, rest time (inactivity greater or equal to 1 s), rest episode count, movement time, movement episode count (separated by rest periods of at least 1 s), stereotypy time, stereotypy episode count (stereotypy behavior greater or equal to 1 s was counted as 1 episode), stereotypy activity count (beam breaks), vertical episode count, vertical activity count (vertical beam breaks), and vertical activity time. Stereotypy behavior was defined as repeatedly breaking the same beam, which typically happens during grooming or head bobbing. Total horizontal distance, total time in stereotypical behavior, and total vertical activity time over the 30-min session were exported and analyzed in Microsoft Excel.

### Mouse treadmill exhaustion assay

A four-lane variable speed mouse treadmill (AccuScan) placed at a 15° angle was used for acclimation and exhaustion testing. The downhill running paradigm was selected over flat or uphill running as it is associated with greater exercise-induced muscle damage, in part, by forcing muscle to lengthen in an eccentric muscle contraction (reviewed by Proske & Morgan [2001]). For acclimation, up to four mice of the same sex were placed in adjacent lanes on the treadmill for 30 min at 0 m/min for 25 min and 3 m/min for 5 min, with the shock pad activated to 0.3 mA on two consecutive days before testing. On the test day with the shock pad activated, mice were placed on the treadmill for a 3-m/min warm-up for 5 min.

Then, the test run was started at a speed of 10 m/min for 5 min, and the speed was increased by 5 m/min every 5 min until reaching a maximum speed of 25 m/min. Mice ran until exhaustion or 15 min at the maximum speed. Exhaustion was defined as a mouse dwell time of over 10 s on the shock pad. Total running time was recorded as soon as the mouse was exhausted. Total running distance was calculated based on the total running time and speed. The mice used for this assay were all 4 mo old.

## Statistical analysis

Statistical analyses were carried out using GraphPad Prism 9 software (GraphPad Software Inc.). Data are shown as the mean ± the standard error of the mean. Two-tailed unpaired $t$ tests or one-way ANOVA with Tukey's post hoc tests was used unless otherwise indicated to determine statistical significance. ns, not significant ($P ≥ 0.05$), $*P < 0.05$, $**P < 0.01$, $***P < 0.001$, and $****P < 0.0001$.

# Supplementary Information

# Acknowledgements

We thank Dr. Julie Nelson at the CTEGD Cytometry Shared Resource Lab for cell sorting, UGA Pathology and KU Pathology core facilities for histological services, and Kristen MacKeil for technical help. R Wang is a recipient of the Kyoto University SPRING Fellowship. This work was supported in part by grants from the American Cancer Society (RSG-1703201-DDC), Japan Society for the Promotion of Science (17K20148 and 21H05048), Takeda Science Foundation, and Uehara Memorial Foundation to T Ito; and the National Institute of Arthritis and Musculoskeletal and Skin Diseases (1R15AR065077-01SA1) to AM Beedle. The authors declare no competing financial interests.

## Author Contributions

R Wang: conceptualization, resources, data curation, formal analysis, validation, investigation, visualization, and writing—original draft, review, and editing.
F Kato: resources, data curation, formal analysis, validation, investigation, and visualization.
RY Watson: resources, data curation, formal analysis, and investigation.
AM Beedle: conceptualization, resources, data curation, formal analysis, investigation, methodology, and writing—review and editing.
JA Call: conceptualization, resources, data curation, formal analysis, investigation, methodology, and writing—review and editing.
Y Tsunoda: methodology.
T Noda: methodology.
T Tsuchiya: software, formal analysis, and writing—review and editing.
M Kashima: software, formal analysis, and writing—review and editing.
A Hattori: conceptualization, resources, formal analysis, and investigation.
T Ito: conceptualization, resources, data curation, formal analysis, supervision, funding acquisition, investigation, project administration, and writing—original draft, review, and editing.

## Conflict of Interest Statement

The authors declare that they have no conflict of interest.

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
