## [Reviewer comments · Life Science Alliance]

Life Science Alliance

The RNA binding protein Msi2 regulates autophagy during myogenic differentiation

Ruochong Wang, Futaba Kato, Rio Watson, Aaron Beedle, Jarrod Call, Yugo Tsunoda, Takeshi Noda, Takaho Tsuchiya, Makoto Kashima, Ayuna Hattori, and Takahiro Ito

DOI: <https://doi.org/10.26508/lsa.202302016>

Corresponding author(s): Takahiro Ito, Kyoto University

Review Timeline:

Submission Date:	2023-02-27
Editorial Decision:	2023-04-03
Revision Received:	2023-12-26
Editorial Decision:	2024-01-25
Revision Received:	2024-01-31
Accepted:	2024-02-05

Transaction Report:

April 3, 2023

Re: Life Science Alliance manuscript #LSA-2023-02016-T

Prof. Takahiro Ito
Kyoto University
Institute for Life and Medical Sciences
Shogoin Kawaharacho 53
Sakyo
Kyoto, Kyoto 6068507
Japan

Dear Dr. Ito,

Thank you for submitting your manuscript entitled "The RNA binding protein Msi2 regulates autophagy during myogenic differentiation" to Life Science Alliance. The manuscript was assessed by expert reviewers, whose comments are appended to this letter. We invite you to submit a revised manuscript addressing the Reviewer comments.

Thank you for this interesting contribution to Life Science Alliance. We are looking forward to receiving your revised manuscript.

Sincerely,

B. MANUSCRIPT ORGANIZATION AND FORMATTING:

Reviewer #1 (Comments to the Authors (Required)):

The manuscript by Wang and colleagues is focused on elucidating the role of the RNA-binding protein Musashi2 (Msi2) in myoblast differentiation and autophagy. Msi2 is a myogenin (Myog) target gene, and is post-transcriptionally regulated during myogenic differentiation. Msi2 knockdown in mouse myoblasts blocked myogenic differentiation without affecting the expression of MyoD or Myog. Msi2 overexpression was sufficient to promote myoblast differentiation and myocyte fusion. Msi2 loss attenuated autophagosome formation via downregulation of ATG8 (LC3) at the early phase of myoblast differentiation. Msi2 knockout mice had smaller limb muscles and poor exercise muscle performance. The authors concluded that Msi2 is a novel regulator of mammalian myogenesis and Msi2 has a functional link between muscular development and autophagy regulation.

Overall, the manuscript is fairly well-written and the experiments are logically designed. Data are supportive of the overall hypothesis presented by the authors. Statistics are fairly strong and are appropriate for the experimental tests used. I have a few questions with regards to some of their interpretations as well as questions as to why specific aspects with regards to the Msi2 KO mouse muscle phenotypes were not fully followed up on and/or pursued. The overall study is of interest to the muscle biology and myogenesis fields; however, more clarifications are warranted.

Major Comments:

1. Myoblast fusion and myoblast differentiation are two different concepts as key markers drive different outcomes. Delayed differentiation (as the Msi2 KO mice are viable) suggests that myogenic differentiation is impaired and fusion is not fully disrupted. If the authors are suggesting myoblast fusion is impaired, known fusion markers (e.g. Myomixer, Bai1) need to be assessed for changes in expression rather than delayed myogenic differentiation as measured via MyHC differentiation/fusion index as performed by the authors. Otherwise, the authors should consider rephrasing the interpretation of some of these findings..
2. I was surprised that no histological assessments of the Msi2 KO mouse muscles was performed. A simple H&E histochemical staining of some of the muscle groups with myofiber cross-sectional area assessments would confirm the smaller myofiber sizes that the authors purported.
3. Similarly to my second point, changes in metabolic function, Msi2 KO muscle performance should be reflected in myofiber type. Histochemical analysis of muscle fiber type (e.g. metachromatic ATPase stain) or immunofluorescent staining of myosin isoforms would confirm a shift in muscle fiber types that might reflect the mechanisms behind the poor exercise performance.
4. With regards to autophagosome formation defects in association with Msi2 knockdown. Have the authors evaluated autophagic flux? Is Lamp2 expression affected in response to Msi2 knockdown in myoblasts?
5. The authors mention that Msi1 was not upregulated in the Msi2 knockdown myoblasts. Was expression changed in the Msi2 KO muscles?

Reviewer #2 (Comments to the Authors (Required)):

Wang et al., investigate the role of the RNA binding protein Msi2 during muscle cell differentiation.

While this topic is interesting and novel, several caveats in this work impede the overall conclusion:

- This work has been done in C2C12, which is not a physiological model. Moreover, the fusion index of the cultures are very low (below 5%) as compared to the literature, strongly suggesting a problem with the cell line used. Therefore, I highly recommend to use culture of primary muscle stem cells.

- Figures 2 & 3 & 6: we can not exclude a role of Msi2 in muscle cell viability. Have you assess cell viability, cell death, apoptosis and senescence in this experimental set-up (i.e. all along muscle cell differentiation)?
- quantifications are not clear : Figures 1E, 2B and 2C. How many biological replicates or independent experiments have been performed? Please show the individual datas.
- Figure 2F: quantification of nuclear staining versus cytoplasm staining would bring important informations
- quantifications are lacking : Figures 2H, 3D, 3E, S3A & S4B, Figures 2E, 3A & 3B for muscle cell size
- Figure 5D: how quantification has been performed? One dot is one cell? Of this is the case, mean of individual cells should be represented.
- Figure 7: How have you validate the efficiency of the deletion of Msi2 in your KO model? Outcomes presented in this figure are not the most relevant to assess the role of Msi2 on adult myogenesis in vivo. Cross-sectional area (CSA), CSA per fiber type, nuclei per fiber, muscle area, number of Pax7+ cells per fiber, number of fibers per muscle must be quantified to have an overview.

Minor comments:

- page 6: the sentence "In contrast..." is repeated twice

List of major changes in the revised version

1. Figure 1A has been replaced with a different replicate.
2. Figures 1E and 2B have been updated to include data from three independent experiments combined, instead of data from a representative experiment.
3. The original Figure 2C was moved to Supplementary Figure S2A, and Figures 2D and E are Figures 2C and D in the current version.
4. New Figure 2E for the mRNA expression analysis of *Myomixer* of control or Msi2 knockdown (KD) C2C12 during differentiation.
5. The original Figures 7C, D, and E were moved to Supplementary Figures S5B, C, and D. Figure 7F is relabeled as Fig. 7C in the revised version.
6. New Figures 7D, E, F and Supplementary Figs. S5E, F, G and H include new data on histological and fiber typing analysis of soleus and TA+EDL muscles from wildtype and Msi2 knockout (KO) mice.
7. New Supplementary Figure S1C on Western blot analysis of Msi family proteins Msi1 and Msi2 in limb muscle samples of Msi2 KO mice.
8. New Supplementary Figure S2C on Western blot analysis of apoptotic regulator proteins PARP and caspase 3 in control and Msi2 KD cells.
9. New Supplementary Figure S4C on expression analysis of LAMP2 of the control or Msi2 KD cells during differentiation.
10. New Supplementary Figure S5A on expression analysis of Msi family proteins Msi1 and Msi2 in Msi2 KO mice.

List of new results included in *Figures for Reviewers*:

11. Figure for Reviewer 1: WB replicates for Fig. 1D.
12. Figure for Reviewer 2: Quantification of nuclear to cytoplasmic Msi2 ratio for Fig. 1F.
13. Figure for Reviewer 3: WB replicates for Fig. 2A.
14. Figure for Reviewer 4: Replicates of Fig. S2A (originally Fig. 2C).
15. Figure for Reviewer 5: Staining of senescent cells during C2C12 differentiation.
16. Figure for Reviewer 6: Cell viability assay during C2C12 differentiation.
17. Figure for Reviewer 7: Quantification of myotube size for Fig. 2D and 3B.
18. Figure for Reviewer 8: WB replicates for Fig. S4B.
19. Figure for Reviewer 9: Fiber size distribution in soleus, TA and EDL muscles.
20. Figure for Reviewer 10: Quantification of nuclei number per fiber in soleus muscles.

Response to Referees' comments

We greatly appreciate the Reviewers' interest in our work and their insightful comments. In response, we have carried out new sets of analysis, and we have now completed most of the experiments suggested by the Reviewers. The data we obtained from these experiments have allowed us to further understand the role of the RNA binding protein Msi2 in myogenic differentiation and skeletal muscle functions. Our point-by-point responses to the Reviewers' comments below, as well as changes and revisions in the manuscript are shown in *blue*. We would be happy to integrate any of the Figures for Referees into Supplementary Figures if the reviewers request.

Reviewer #1:

The manuscript by Wang and colleagues is focused on elucidating the role of the RNA-binding protein Musashi2 (Msi2) in myoblast differentiation and autophagy. Msi2 is a myogenin (Myog) target gene, and is post-transcriptionally regulated during myogenic differentiation. Msi2 knockdown in mouse myoblasts blocked myogenic differentiation without affecting the expression of MyoD or Myog. Msi2 overexpression was sufficient to promote myoblast differentiation and myocyte fusion. Msi2 loss attenuated autophagosome formation via downregulation of ATG8 (LC3) at the early phase of myoblast differentiation. Msi2 knockout mice had smaller limb muscles and poor exercise muscle performance. The authors concluded that Msi2 is a novel regulator of mammalian myogenesis and Msi2 has a functional link between muscular development and autophagy regulation.

Overall, the manuscript is fairly well-written and the experiments are logically designed. Data are supportive of the overall hypothesis presented by the authors. Statistics are fairly strong and are appropriate for the experimental tests used. I have a few questions with regards to some of their interpretations as well as questions as to why specific aspects with regards to the Msi2 KO mouse muscle phenotypes were not fully followed up on and/or pursued. The overall study is of interest to the muscle biology and myogenesis fields; however, more clarifications are warranted.

We very much appreciate the reviewer's high evaluation of our study and many constructive comments. Below we describe the experiments we have carried out to directly address the reviewer's comments. In addition, we agree that the muscle phenotypes of the Msi2 mutant mice should also be examined and thus described in the revised version. We performed analysis on muscle histology and fiber types. The data are added as new Figures 7D, E, F, and Supplementary Figures 5E, F, G, H.

Major Comments:

1. Myoblast fusion and myoblast differentiation are two different concepts as key markers drive different outcomes. Delayed differentiation (as the Msi2 KO mice are viable) suggests that myogenic differentiation is impaired and fusion is not fully disrupted. If the authors are suggesting myoblast fusion is impaired, known fusion markers (e.g. Myomixer, Bai1) need to be

assessed for changes in expression rather than delayed myogenic differentiation as measured via MyHC differentiation/fusion index as performed by the authors. Otherwise, the authors should consider rephrasing the interpretation of some of these findings..

We thank the reviewer's comment. To answer the comment, we have performed expression analysis on the fusion genes Myomixer and Bai1. As in our new results shown in Fig. 2E, the expression levels of Myomixer increased in control cells during differentiation, but slower in Msi2 KD cells. This result suggests myoblast fusion is also impaired by the Msi2 loss.

We also analyzed *Bai1* mRNA expression by RT-qPCR, and found that it is under detectable level in C2C12 cells during differentiation. The level is even underdetected in our C2C12 positive control sample, which is a very concentrated RNA sample collected from differentiated C2C12 myotubes.

2. I was surprised that no histological assessments of the Msi2 KO mouse muscles was performed. A simple H&E histochemical staining of some of the muscle groups with myofiber cross-sectional area assessments would confirm the smaller myofiber sizes that the authors purported.

In response to the reviewer's suggestion, we have performed histological assessments using H&E staining of soleus, tibialis anterior (TA), and extensor digitorum longus (EDL) muscles from control and Msi2 KO mice (Figs. 7D and S5E). The total cross-sectional area, fiber number and average fiber size are all decreased in Msi2 KO mice compared with those in WT mice (Supplementary Figs. S5F and G). These data demonstrated that Msi2 loss impaired myofiber formation and muscle development *in vivo*.

3. Similarly to my second point, changes in metabolic function, Msi2 KO muscle performance should be reflected in myofiber type. Histochemical analysis of muscle fiber type (e.g. metachromatic ATPase stain) or immunofluorescent staining of myosin isoforms would confirm a shift in muscle fiber types that might reflect the mechanisms behind the poor exercise performance.

We agree this is an important question. We therefore performed immunofluorescent staining of myosin heavy chain (MHC) isoforms to compare the fiber compositions in soleus, TA and EDL muscles between wildtype (WT) control and Msi2 KO mice (Figs. 7E and F). We observed a change in the fiber composition, and surprisingly, there were more oxidative fibers at the expense of mixed or more glycolytic fibers. Soleus muscles consisted of 43% of type I oxidative slow fibers in KO tissues while 34% of type I fibers in WT (Supplementary Fig. S5H). In TA and EDL muscles, type IIb glycolytic fast fibers were decreased to 74% in Msi2 KO tissues compared with 83% in WT. Moreover, Msi2 KO TA and EDL muscles contained 4% of type I oxidative fibers, which were typically not detected in TA and EDL from WT controls.

Slow oxidative fibers play an important role in endurance exercise. Msi2 KO mice exhibit significantly diminished endurance capacity despite their muscles having more type I fibers. It is

possible that the overall reduction in the muscle size, fiber numbers and fiber sizes may be more critical and lead to the impaired exercise performance in Msi2 KO mice. Another possibility is that these phenotypically “type I slow fibers” in KO muscles may promiscuously express MHC I protein and may not function as slow oxidative fibers. Future studies will be necessary to understand mitochondrial oxidative capacity and metabolic functions in Msi2 KO skeletal muscles.

4. With regards to autophagosome formation defects in association with Msi2 knockdown. Have the authors evaluated autophagic flux? Is Lamp2 expression affected in response to Msi2 knockdown in myoblasts?

In Fig. S4A, we have assessed autophagic flux using Bafilomycin treatment, which can block the breakdown of autophagosome and LC3 proteins. As quantified in Fig. S4B, the autophagic flux, measured by the differences in LC3B-II levels with or without Bafilomycin treatment, was decreased by Msi2 knockdown, which was rescued by Msi2 overexpression. And as per the reviewer’s suggestion, we have added data on LAMP2 expression (shown in Supplementary Fig. S4C). These results indicated that the LAMP2 level increased during C2C12 differentiation, but was not affected by Msi2 knockdown, suggesting Msi2 regulates autophagosome formation but not lysosome biogenesis.

5. The authors mention that Msi1 was not upregulated in the Msi2 knockdown myoblasts. Was expression changed in the Msi2 KO muscles?

In response to this question, we performed WB analysis of Msi1 and Msi2 proteins in Msi2 KO mouse muscles. Results are included in Supplementary Fig. S5A. We found that Msi2 expression is almost depleted, as expected, in soleus and gastrocnemius muscles from the Msi2 KO mice, while the level of Msi1 protein was comparable between WT control and Msi2 KO muscles. This result suggests Msi2 KO does not affect the expression of Msi1 in muscles.

Reviewer #2:

Wang et al., investigate the role of the RNA binding protein Msi2 during muscle cell differentiation.

While this topic is interesting and novel, several caveats in this work impede the overall conclusion:

We greatly appreciate the reviewer’s interest in our work and insightful comments. As detailed in the responses below, we have carried out multiple key experiments to address the reviewer’s concerns about our model.

- This work has been done in C2C12, which is not a physiological model. Moreover, the fusion index of the cultures are very low (below 5%) as compared to the literature, strongly suggesting a problem with the cell line used. Therefore, I highly recommend to use culture of primary muscle stem cells.

We thank the reviewer for their comment. First, we would like to clear a misunderstanding about the fusion index. In some literature, the fusion index is defined as the proportion of nuclei in MHC-positive cells with respect to the total number of nuclei in a field of view, which we named as “differentiation index” in this manuscript. As shown in Figure. 2F, the differentiation index of our control cells is over 30%, which is comparable to those in published literature. The fusion index we have shown here is defined as the average number of nuclei in MHC-positive cells in a field of view. We believe this reflects the extent of how many times the cells undergo fusion. The definitions of both indices were described in the result section when they were first mentioned.

Here we would like to cite several prior studies that use C2C12 cells as well (Bajaj et al., 2011; Wang et al., 2019. Nie et al., 2020). The differentiation indices in these publications range from 20 to 30%, which is pretty much equivalent to 30% we observed in our experiments using C2C12 cells. The results indicate that the differentiation ability of our C2C12 cell line is comparable to those in published literatures. These facts would justify the use of C2C12 as one of the models for myogenic differentiation and our data, at least in part, are valid. Moreover, according to the reviewer’s suggestions, now we have more *in vivo* data confirming our findings in C2C12, as shown in Figure 7 and Supplementary Figure S5.

- Figures 2 & 3 & 6: we can not exclude a role of Msi2 in muscle cell viability. Have you assess cell viability, cell death, apoptosis and senescence in this experimental set-up (i.e. all along muscle cell differentiation)?

As suggested by the reviewer, and in addition to the cell growth curve analysis shown in Supplementary Fig. S2A in the revised version (originally in Fig. 2C), we analyzed apoptosis by testing caspase3 and PARP cleavages in immunoblots (Supplementary Fig.S2C). We also performed senescence-associated beta-galactosidase (SA- β -gal) staining to assess cellular senescence (Figures for Reviewers 5), and cell viability analysis by using trypan blue staining (Figures for Reviewers 6), all along control and Msi2 KD C2C12 cells during differentiation. As shown in these figures, we found caspase 3 and PARP protein cleavages, and trypan blue-negative live cell population were all comparable between the control and Msi2 KD cells along the differentiation process. These results demonstrate that Msi2 loss does not induce apoptosis in myoblast or affect cell viability during differentiation. By quantification, SA- β -gal-positive cell area was also comparable between the control and KD cells on differentiation days 0, 1, and 5, and is lower in Msi2 KD cells on day 3. This result indicates that Msi2 KD may not affect the senescence process during myoblast differentiation.

- *quantifications are not clear : Figures 1E, 2B and 2C. How many biological replicates or independent experiments have been performed? Please show the individual datas.*

In the original submission, data in Figures 1E and 2B were from a representative experiment. In the revised version, we have included all the data from three independent experiments. If the reviewer is interested, we have included individual data from three replicates in the Figures for Reviewers for their review (Figures for Reviewers 1 and 3).

The original Figure 2C contained data from two independent experiments, with two replicates in each experiment, as described in the figure legend. We added all these individual data as Figure 4 in the Figures for Reviewers for reference.

- *Figure 2F: quantification of nuclear staining versus cytoplasm staining would bring important informations*

We guess the reviewer meant Figure 1F, not Figure 2F. As suggested, we quantified the staining of Msi2 in nuclei versus cytoplasm, and the data are shown in Fig. 2 in the Figures for Reviewers. The result shows no significant differences in the nuclear to cytoplasmic Msi2 ratio during differentiation.

- *quantifications are lacking : Figures 2H, 3D, 3E, S3A & S4B, Figures 2E, 3A & 3B for muscle cell size*

Figures 2H, 3E, S3A are representative images showing the mitochondria morphology. We have not performed any quantification on mitochondrial morphology because we did not claim any changes in quantitative traits or properties in our work. It would also be difficult to accurately quantify the morphological changes due to technical limitations. Electron microscopy analysis was performed with sections of epon-embedded cell pellets, and we could not control the direction of how cells were being sectioned. Nonetheless, as the reviewer pointed out, this is an important point. Future studies will be necessary to assess the role of Msi2 in mitochondrial morphology and functions.

Figure S4B shows the quantification of autophagy flux shown in Figure S4A. For reference, we have included results from three other independent experiments as Figures for Reviewers 8, which also supported the original results that LC3 flux is regulated by Msi2.

In Figure 3d, we did not quantify any cell shape parameters in phase contrast images because it is not necessary and we simply wish to present representative images showing more and larger myotubes in the Msi2 OE group. Instead, we would like to provide quantification of MHC+ cells in immunofluorescence staining results from Figures 3B, as we discuss below.

Regarding cell size quantification in Figures 2E, 3A, and 3B in the original submission, we have observed a significantly lower number of MHC+ cells, differentiation, and fusion indices in Msi2 KD cells compared to control cells. It is technically challenging to accurately measure the sizes of myotubes; accurate quantification of myotube lengths is difficult because few myotubes can

be fully captured in an image. The myotube widths are hard to determine since some myotubes are not completely tube-shaped. Nonetheless, in order to answer the reviewer's points, we quantified myotube sizes by measuring fluorescently stained areas in the immunofluorescence staining of MHC protein in the original Figures 2E and 3B (Figs. 2D and 3B in the current submission). As shown in Fig. 7 of Figures for Reviewers, myocytes are smaller in their sizes in the absence of Msi2.

- Figure 5D: how quantification has been performed? One dot is one cell? Of this is the case, mean of individual cells should be represented.

Yes, each dot represents one cell that has been analyzed. And more than 30 cells from 3 different fields were analyzed in each group, as described in the figure legend. Due to our limited understanding of the question, we may not provide responses on the "mean of individual cells", as requested by the reviewer. For reference, the average puncta number of LC3A is 17.6 for the control and 2.7 for the Msi2 KD cells. For LC3B, that is 16.1 for the control and 6.7 for the Msi2 KD cells. We hope these results answer the question.

- Figure 7: How have you validate the efficiency of the deletion of Msi2 in your KO model? Outcomes presented in this figure are not the most relevant to assess the role of Msi2 on adult myogenesis in vivo. Cross-sectional area (CSA), CSA per fiber type, nuclei per fiber, muscle area, number of Pax7+ cells per fiber, number of fibers per muscle must be quantified to have an overview.

We agree that these points raised are important. In the revised version, we have added the immunoblot result showing the expression of Msi2 is lost in skeletal muscles (Supplementary Fig. S5A). We also performed histological analysis of soleus, TA and EDL muscles, and results presented in Fig. 7D and Supplementary Fig. S5E. We measured the cross-sectional area of the muscles, the number of fibers per muscle, and the average fiber size (Supplementary Figs. S5F and G). We found that Msi2 KO mice have smaller muscles with reduced fiber numbers and fiber sizes. Regarding the CSA per fiber type, we performed immunostaining of MHC isoforms and analyzed the fiber size distribution of major fibers as type I, type IIa in soleus, and type IIb fibers in TA + EDL muscles (Figures for Reviewer 9). We found that both type I and IIa fibers in KO soleus are smaller than those in WT, which is consistent with the smaller average fiber size shown in Supplementary Fig. S5F. We observed the same pattern in type IIb fiber in TA and EDL muscles as well. We also compared the fiber type distribution in the soleus, as well as TA and EDL muscles, between WT and Msi2 KO mice. From this analysis, we found that Msi2 KO mice exhibit a fiber type shifting from fast glycolytic fibers to MHC I-positive slow fibers as shown in Figure 7F. Regarding the number of nuclei per fiber, we found that they are comparable between the WT and KO soleus muscles (Figures for Reviewer 10). We did not notice any other apparent abnormalities in Msi2 KO muscle by the histological and fiber type analyses, such as changes in numbers of muscle fibers with central nuclei.

Minor comments:

- page 6: the sentence "In contrast..." is repeated twice

Thank you very much for noticing the mistake and we have corrected it in the revised manuscript.

References cited in the response letter

Bajaj, Piyush, *et al.* "Patterning the differentiation of C2C12 skeletal myoblasts." ***Integrative Biology*** 3.9 (2011): 897-909.

Wang, YuXin, *et al.* "SPARCL1 promotes C2C12 cell differentiation via BMP7-mediated BMP/TGF- β cell signaling pathway." ***Cell Death & Disease*** 10.11 (2019): 852.

Nie, Yaping, *et al.* "Zfp422 promotes skeletal muscle differentiation by regulating EphA7 to induce appropriate myoblast apoptosis." ***Cell Death & Differentiation*** 27.5 (2020): 1644-1659.

Fig. 1

Original in Fig. 1D

Replicates of Fig. 1D

Figure 1. Independent replicates of WB shown in Fig. 1D.

Fig. 2
Quantification of Fig. 1F

Figure 2. Quantification of nuclear Msi2 to cytoplasmic Msi2 ratio during C2C12 differentiation. Five independent fields from each time point as shown in Fig. 1F were used for analysis.

Fig. 3
Original in Fig.2A

Replicates of Fig. 2A

Figure 3. Independent replicates of WB shown in Fig. 2A.

Fig. 4 Replicates of Fig. S2A

Figure 4. The two independent replicates of cell growth assay which are combined as Fig. S2A.

Fig. 5 SA-b-gal staining

Figure 5. Senescent-associated beta-galactosidase (SA-b-gal) staining of C2C12 cells during differentiation. SA-b-gal positive cells were considered as senescent cells and stained with blue color. Both shLuc and shMsi2 showed minimal number of blue cells at early stage of differentiation. The portion of SA-b-gal positive cells was lower in Msi2 KD cells on day3. Cells in positive control were treated with 1 μ M staurosporine for 3 hours before differentiation.

Fig. 6 Cell viability assay

Figure 6. Cell viability assay of control and Msi2 KD C2C12 cells during differentiation. Cells were trypsinized, stained with Trypan blue and counted at indicated time during differentiation. Trypan blue positive cells were counted as dead cells. Very little portion of cells were observed dead in both control and Msi2 KD cells, suggesting Msi2 absence does not impair cell viability. Three independent replicates were included in each group.

Fig. 7 Quantification of C2C12 myotube size

Figure 7. Quantification of myotube size in data from Fig. 2D and Fig. 3B. Five independent fields were evaluated in each group.

Fig. 8

Original in Fig. S4B

Replicates of Fig. S4B

Figure 8. Independent replicates of WB in Fig. S4B. Cells were treated with DMSO control (-) or Bafilomycin (+) for 3 hours or as indicated before protein collection.

Fig. 9

Figure 9. Fiber size distribution of type I, type IIa fibers in soleus and type IIb fibers in TA and EDL muscles. Two individual mice were included in each group. Muscles were dissected from two hindlimbs of two mice, and four individual sections were stained for MHC expression for fiber typing.

Fig. 10

Figure 10. Quantification of soleus muscles nuclei number per fiber. Data was acquired from hematoxylin and eosin stained sections as Fig. 7D. Four individual sections from two solei from two individual mice were included in each group.

January 25, 2024

RE: Life Science Alliance Manuscript #LSA-2023-02016-TR

Prof. Takahiro Ito
Kyoto University
Institute for Life and Medical Sciences
Shogoin Kawaharacho 53
Sakyo
Kyoto, Japan 6068507

Dear Dr. Ito,

Thank you for submitting your revised manuscript entitled "The RNA binding protein Msi2 regulates autophagy during myogenic differentiation". We would be happy to publish your paper in Life Science Alliance pending final revisions necessary to meet our formatting guidelines.

- please be sure that the authorship listing and order is correct
- please add the Twitter handle of your host institute/organization as well as your own or/and one of the authors in our system
- please consult our manuscript preparation guidelines <https://www.life-science-alliance.org/manuscript-prep> and make sure your manuscript sections are in the correct order and labeled correctly (Experimental Procedures should be labeled as Materials & Methods...)
- please move your main and supplementary figure legends to the main manuscript text after the references section
- figure S3 has only one panel, so there is no need to label it as A. Please correct this in the figure, its legend and callout in the manuscript file
- please use the [10 author names et al.] format in your references (i.e., limit the author names to the first 10)
- please revise the legend for Figures 6 and S5 so that the panels are introduced in order
- please add a conflict of interest statement to your main manuscript text

A. FINAL FILES:

B. MANUSCRIPT ORGANIZATION AND FORMATTING:

Sincerely,

Reviewer #1 (Comments to the Authors (Required)):

The authors have made significant improvements to the manuscript in demonstrating that the RNA-binding protein MSI2 is a regulator of myogenesis and autophagy. Through analysis of the Msi2 KO mice, the authors demonstrate that the Msi2 KO mice have smaller myofibers, decreased myoblast differentiation/fusion rates, and downregulated autophagy via ATG8/LC3. The authors have provided supportive data in addressing my previous critiques with regards to addressing issues in relation to assessing autophagic flux using Bafilomycin treatment after Msi2 knockdown in muscle cells. LAMP2 increased, suggesting that lysosomal activity was independent of autophagic processes. The statistical analysis is supportive and the experiments are logical. The reported fiber type differences in the Msi2 KO mice are interesting, but are logical based on the reported differences in both cross-sectional area and coloration shown in Fig. 7B. I believe this manuscript will be of interest to the muscle autophagy and myogenic differentiation fields. I have no further issues with this manuscript.

February 5, 2024

RE: Life Science Alliance Manuscript #LSA-2023-02016-TRR

Prof. Takahiro Ito
Kyoto University
Institute for Life and Medical Sciences
Shogoin Kawaharacho 53
Sakyo
Kyoto, Japan

Dear Dr. Ito,

Thank you for submitting your Research Article entitled "The RNA binding protein Msi2 regulates autophagy during myogenic differentiation". It is a pleasure to let you know that your manuscript is now accepted for publication in Life Science Alliance. Congratulations on this interesting work.

DISTRIBUTION OF MATERIALS:

Again, congratulations on a very nice paper. I hope you found the review process to be constructive and are pleased with how the manuscript was handled editorially. We look forward to future exciting submissions from your lab.

Sincerely,
